# A candidate sex determination locus in amphibians which evolved by structural variation between X- and Y-chromosomes

Heiner Kuhl [1,13], Wen Hui Tan [2,13], Christophe Klopp [3], Wibke Kleiner [1], Baturalp Koyun [1,4], Mitica Ciorpac [5,6], Romain Feron [7,8], Martin Knytl [9,10], Werner Kloas [1], Manfred Schartl [11,12], Christoph Winkler [2] ✉ & Matthias Stöck [1] ✉

Most vertebrates develop distinct females and males, where sex is determined by repeatedly evolved environmental or genetic triggers. Undifferentiated sex chromosomes and large genomes have caused major knowledge gaps in amphibians. Only a single master sex-determining gene, the *dmrt1*-paralogue (*dm-w*) of female-heterogametic clawed frogs (*Xenopus*; ZW♀/ZZ♂), is known across >8740 species of amphibians. In this study, by combining chromosome-scale female and male genomes of a non-model amphibian, the European green toad, *Bufo*(*tes*) *viridis*, with ddRAD- and whole genome pool-sequencing, we reveal a candidate master locus, governing a male-heterogametic system (XX♀/XY♂). Targeted sequencing across multiple taxa uncovered structural X/Y-variation in the 5′-regulatory region of the gene *bod1l*, where a Y-specific non-coding RNA (ncRNA-Y), only expressed in males, suggests that this locus initiates sex-specific differentiation. Developmental transcriptomes and RNA in-situ hybridization show timely and spatially relevant sex-specific ncRNA-Y and *bod1l*-gene expression in primordial gonads. This coincided with differential *H3K4me*-methylation in pre-granulosa/pre-Sertoli cells, pointing to a specific mechanism of amphibian sex determination.

Most vertebrates occur as females and males. Vertebrate sex determination comprises environmental triggers, or simple to complex genetic (genotypic) sex determination—and perhaps combinations thereof—that evolved several times[1,2]. Likewise, the genetic and cellular developmental processes directing undifferentiated gonads towards testes or ovaries are diverse[3].

Genetic sex determination is governed by sex chromosomes that are typically differentiated and often microscopically distinguishable in mammals, birds, and several reptile groups[4]. With several exceptions, some reptiles and most other poikilothermic vertebrates, like fish and amphibians, exhibit poorly or undifferentiated (homomorphic) sex chromosomes, which are prevailing in the tree of life[5]. In vertebrates, three main evolutionary concepts explain this prevalence:

(i) frequent turnovers, overthrowing sex-determining genes by novel autosomal mutations, or even the sex determination system (e.g., XY to WZ)[6], (ii) recurrent X-Y recombination, e.g., by occasional sex reversal to XY-females[7], and (iii) "jumping master genes" with conserved genetic triggers translocating to other chromosomes[8]. Conserved poorly or non-differentiated sex chromosomes are known in several clades of poikilothermic vertebrates, e.g., in sturgeons, pikes, and skinks[9–11].

The majority of amphibians possess undifferentiated sex chromosomes with either female (ZZ/ZW) or male (XY/XX) heterogamety[12,13]. Several examples of cytogenetically differentiated sex chromosomes[13,14], a female W0/00 male sex determination system[15], and several amphibians with multiple sex chromosomes have

been identified in both, laboratory strains[16] and natural populations[17]. Notably, to date, in >8740 species of amphibians[18], the widespread combination of poorly or undifferentiated sex chromosomes with large genome-sizes (see below) have caused a basic knowledge gap with only a single though very well-characterized amphibian master sex-determination gene known in a model species, the African clawed frog (*Xenopus laevis*[12,19]) and some related species[20]. The *Dm-w* gene arose after (and perhaps in response to) tetraploidization, is essential as recently shown in knock-out females[21], and has been lost secondarily in some tetraploid species[22].

True toads, Bufonidae, share large genomes, conserved karyotypes[23] and undifferentiated sex chromosomes[24]. Typical of most bufonid toads is also Bidder's organ that develops in males and - in many species—also in females during early gonadogenesis from the anterior parts of the primordial gonads, considered to possibly present a rudimentary ovary[25]. Using sex-reversal experiments, castrating adult males and regrowing functional ovaries from Bidder's organs by Harms[26], early on, Ponse[27] inferred female heterogamety (ZW) in common toads (*B. bufo*). This was also cytogenetically deduced in related *B. gargarizans*[28], and cane toads (*Rhinella marina*[29]). With few exceptions, a meta-analysis of sex-differential F1-hybrid fitness of interspecific crosses in bufonids likewise supported widespread female heterogamety[30].

This situation contrasts with that in our target species, Palearctic green toads of the *Bufo(tes) viridis* sub-group. While their sex chromosomes are also undifferentiated[24,31], sex-reversal experiments in 'B. viridis' from Asia Minor[32] suggested an XY-system. Molecular evidence

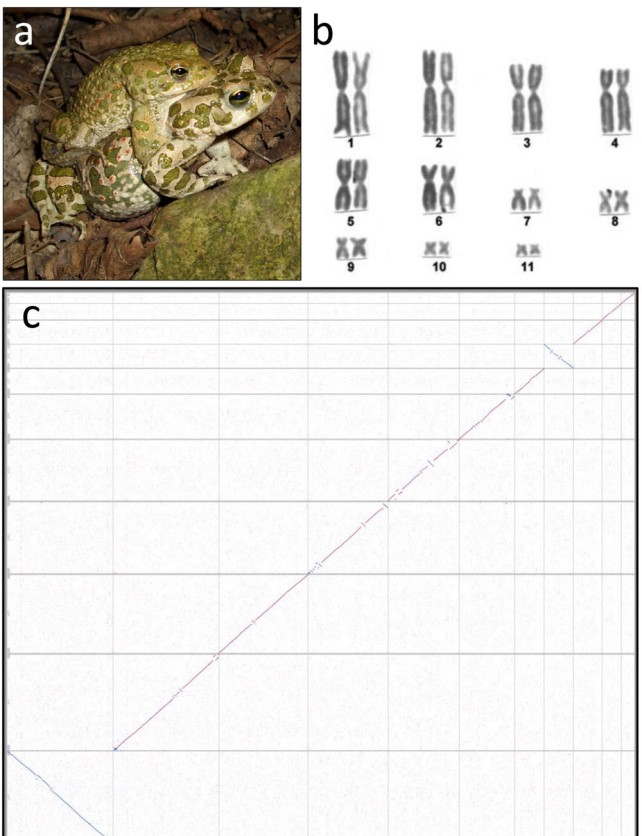

**Fig. 1 | The chromosome-scale reference-genome of *Bufo(tes) viridis*, its karyotype and comparison with the reference genome of *Bufo bufo*. a** Breeding pair of *B. viridis* in amplexus (© M. Stöck). **b** Giemsa-stained karyotype of *B. viridis* (© M. Stöck). **c** *B. viridis*-assembly (y-axis) compared to *B. bufo* (x-axis) assembly by the Vertebrate Genome Project (VGP, GenBank accession number GCA_905171765.1) showing high synteny of both bufonid genomes across all 11 chromosomes.

for male heterogamety from sibship analyses using microsatellites and nuclear sequence markers[33–35], identified the largest linkage group (LG1) as sex-linked in several diploid green toad species. This linkage group is homologous to the autosomal LG1 of *X. tropicalis* and harbors the gene *dmrt1*. All the green toad species feature strongly-reduced X/ Y-recombination but show some X/Y-exchange in the pseudo-autosomal region over evolutionary time scales[33]. However, phylogenetic analysis revealed X- and Y-alleles of *dmrt1* to cluster by species and not by gametologue, suggesting X/Y-recombination involving *dmrt1*, which rejected its role as master sex determiner[35]. Details of sex determination have not been examined in allopolyploid green toads[36].

In mammals (6718 species[37]), genome assemblies are available for about half of all species (>3150[38]). Although there are more species described in amphibians (>8740[18]), owing to large genome sizes and excessive repetitive regions, to date only >160 amphibian genomes have been assembled, including 95 out of 7705 Anura, 60 out of 816 Urodela (Caudata) and 7 out of 222 Gymnophiona[18,38]. However, profiting from long-read- and 3D-sequencing technologies, amphibian genomics is on the rise.

Here, we provide a female and a male chromosome-scale genome of a non-model amphibian, the European green toad, *Bufo(tes) viridis*. We identify a specific candidate master sex determining locus - present in multiple related green toad taxa (namely: *balearicus, shaartusiensis, siculus, turanensis, variabilis*)[36] - that evolved by structural variation, involving a Y-specific ncRNA, in the regulatory region of a gene (*bod1l*) between X and Y chromosomes.

## Results

### Chromosome-scale genome assembly of *B. viridis*

We obtained a 3.8 Gbp chromosomal-level genome assembly using Oxford Nanopore Technologies (ONT) long-read (N50 = 11.6 kbp; sequencing coverage 21x) and Illumina short-read data (2 × 150 bp; sequence coverage 47x) for the European green toad (Fig. 1a). N50 contig size of the final assembly was 1.9 Mbp. Eleven chromosomes (scaffold N50 = 470 Mbp; consisting of 94% of the contigs' total length), consistent with the karyotype of *B. viridis* (Fig. 1b), were constructed using chromosome conformation capture (Hi-C) short reads (318 Gbp). Many unplaced contigs contain a known long satellite DNA family[39]. Comparing the chromosomes to the *B. bufo* assembly revealed genome-wide synteny and collinearity with few inversions, despite significant differences in assembly size (VGP's aBufBuf1.1 = 5.0 Gbp; Fig. 1c). De novo analysis revealed 68.5% repeat content in *B. viridis*. Using published data from related species and RNAseq data, we annotated 21,605 protein coding genes in the genome. BUSCO (Geneset vertebrata_ODB9)-scoring of protein predictions revealed 96.4% complete coding sequences and a low missing gene fraction of 1.7% (C:96.4% [S:94.2%, D:2.2%], F:1.9%, M:1.7%, n = 2586).

### ddRADseq in *B. viridis* genetic families

We first attempted to identify the sex determination region by double digest restriction-site associated DNA sequencing (ddRADseq). In three genetic families of European green toads (one comprising parents and its 53 offspring, two other groups of 24 and 15 siblings from unknown parents), we analyzed a total of 443,378 markers, present in at least one individual with a depth higher than 4. Of these, 96 were significantly associated with male sex ($p < 0.05$; Bonferroni correction), but none was found in all males (Supplementary Fig. 1), probably due to high stochasticity of random sequencing of large genome fractions. When mapped to the *B. viridis* genome, 64 of these 96 markers were located along the entire chromosome 1 (scf1) (Supplementary Fig. 2), one on chromosome 7 (scf7), five on unplaced scaffolds, and 26 were not aligned uniquely or with low quality (mapping quality, MQ < 20). Overall, this strongly supports a male (XY) genetic sex determination system on chromosome 1, however, with a very small sex-specific region that is not fully covered in each individual,

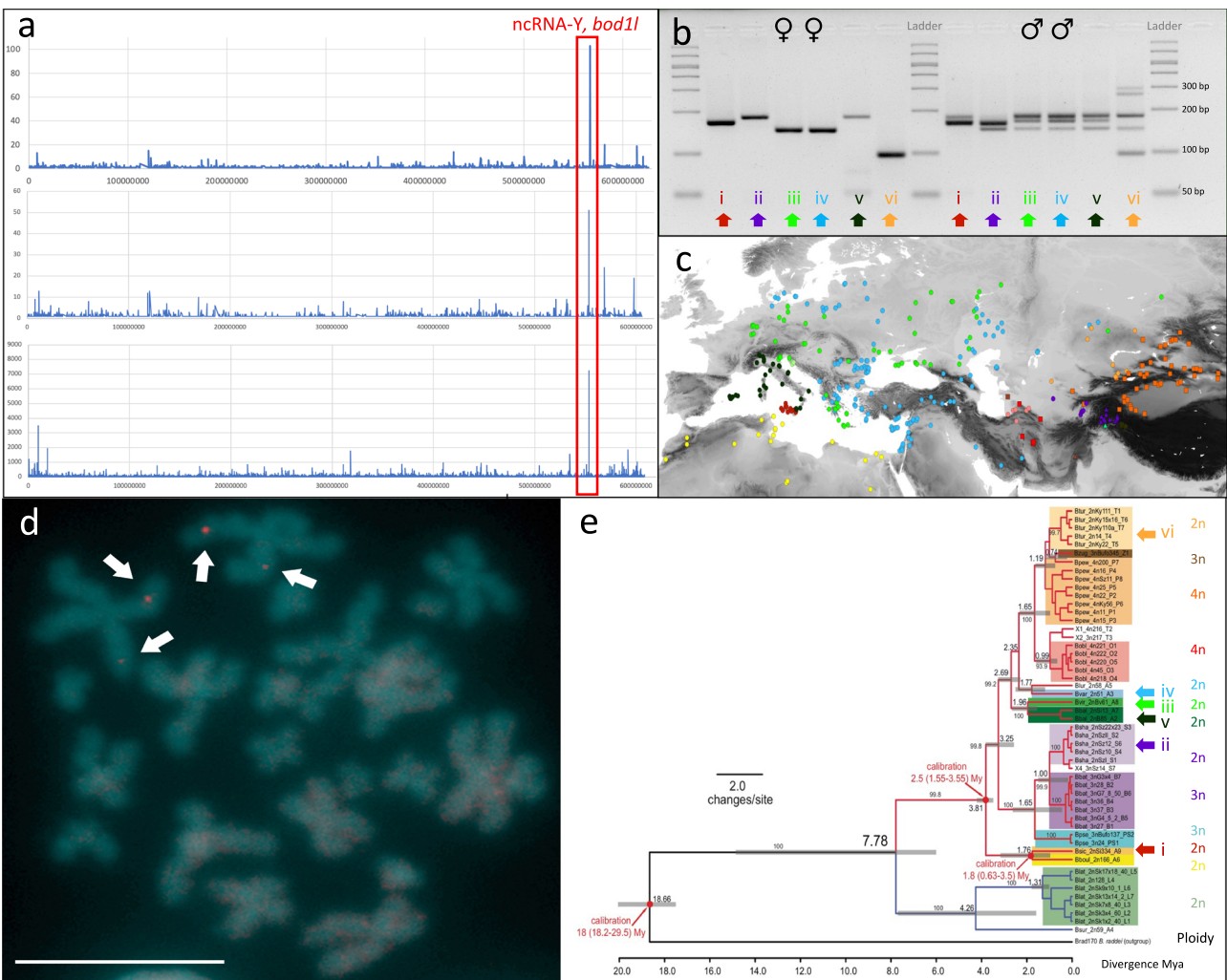

**Fig. 2 | The sex-diagnostic marker of *Bufo(tes) viridis* and its diploid relatives among Palearctic green toads. a** top: Mapping the pool-seq short reads to the *B. viridis* reference showed genome-wide a single region on scaffold 1, where the number of heterozygous SNPs (*y*-axis) (*N* = 103) in males surpassed about nine times the strongest signal of female heterozygous SNPs (*N* = 12), provides strong evidence for male heterogamety (XX/XY; for a putative ZZ/ZW system: Supplementary Fig. 3); **a** middle: male heterozygous SNPs on scaffold 1 of the male genome assembly identifies the same region (note a slight shift of the peak showing the same locus due to different reference genomes); **a** bottom: male-specific pool-seq ('pool-sex') coverage on the male genome (for the entire female and male genomes: Supplementary Data 6); *y*-axis for heterozygous SNPs are counts of SNPs (above and middle subfigure) or Y-specifically pool-sex covered base pairs (lower subfigure) in a cluster, where the maximal distance of variants is 9000 bp.
**b** Representative sex-specific PCR-products of primers BvXY1 in females (♀♀, left) and males (♂♂, right) of six (i-vi) nominal diploid green toad taxa, electrophoresis result using 4 μL of PCR-products on a 4% high-resolution Sieve-agarose gel run for

60 min on 70 V; examined numbers of females (♀) and males (♂) for Palearctic green toad taxa were: (i) *siculus* (4♀, 6♂), (ii) *shaartusiensis* (highest band shown in males is weak; 4♀, 5♂), (iii) *viridis* and (iv) *variabilis* (49♀, 44♂), (v) *balearicus* (6♀, 4♂), (vi) *turanensis* (6♀, 4♂). **c** Range map of Palearctic green toad taxa (based on mitochondrial DNA and ploidy inferred from karyotypes, flow cytometry and nuclear microsatellite genotypes), numbers (i-vi) and color-coding as in (**b**) and (**e**), diploid (2n) taxa are indicated by circles, triploids (3n) by triangles and tetraploids (4n) by squares (map ca. 11° W to 97° E, 25° N to 62° N). **d** FISH-TSA on a metaphase spread from a *B. viridis* male with red *bod1l*-signals on chromosomes 1 on their short (p) arm (arrows), chromosomes were counterstained with DAPI (blue-green), scale bar in lower left corner represents 10 μm (ca. 40 metaphases were inspected).
**e** Dated mitochondrial phylogeny of Palearctic green toads (adapted from ref. 36, with the phylogenetic positions of the diploid taxa, in which BvXY1 is sex-diagnostic, indicated by color-coded arrows as in (**b**, **c**); the three ploidy levels are provided to the right of the leaves, note that the sex marker is seemingly conserved only in the diploid green toads; all polyploids are of hybrid origin[36].

and sex-linkage of most ddRAD markers is not evolutionarily conserved due to X-Y recombination, in the huge pseudo-autosomal region (PAR) of this chromosome 1. While this approach powerfully corroborated previous microsatellite-based results[33], it did not provide a sex-diagnostic marker.

### Pool-seq yields a cross-amplifying marker in multiple taxa

To narrow down the sex determining region on scaffold 1, we used whole genome pool sequencing (from here: "pool-sex"), involving 22 female and 22 male phenotypically-sexed adult green toads from multiple European green toad populations (Supplementary Data 1), chosen to be closely related according to phylogeographic

information[40,41] and presumably sharing the same sex chromosome, based on microsatellites[33–35]. Mapping pool-sex short reads to the reference genome revealed a single peak in a region on scaffold 1 (scf1), where the number of clustered heterozygous SNPs (*N* = 103) in males surpassed about nine times the maximum number of clustered heterozygous SNPs (*N* = 12) in females (Fig. 2a, Supplementary Data 2; Supplementary Fig. 3). This result was later confirmed quantifying male-heterozygous SNPs on scaffold 1 of the male genome assembly as well as male-specific pool-seq coverage on the male genome (see below and Fig. 2a).

This supports a male-heterogametic (XY) system, involving a short 66 kb region (scf1:566,783,058-566,848,882 bp). This locus

encompasses one third of the 5′-upstream-region and two thirds of the gene *bod1l* (bufVir1.4895.2, biorientation of chromosomes in cell division 1 like 1). Transcriptomes of 54 larval, juvenile and adult toads (see below) and cDNA-based PCR-products of males of *B. viridis* exhibited consistent heterozygosity of exons 7 to 10 (scf1:566,831,904-566,833,300), while females were always homozygous.

The heterozygous 5′-region exhibited several INDELs between X and Y haplotypes, enabling primer design. In females, primers BvXY1 amplified a single PCR-product—visible as one band on the gel, and two to three bands (five in *B. turanensis*) in males, widely consistent with an XX♀/XY♂-system, but suggesting multiple Y-specific signals in males, which might be sex-chromosomal sequence duplications, nonspecific PCR-amplicons or even PCR-recombinants. Of note, BvXY1-primers amplified not only sex-specific products in individual *B. viridis* (*sensu lato*) males and females of the pool-sex-approach but also in multiple phenotypically-sexed males and females of this and four additional taxa (*B. balearicus*, *B. siculus*, *B. turanensis*, *B. shaartusiensis*), comprising a monophyletic radiation, probably >3 Mio. years old[36] (Fig. 2b–e). The sex-diagnostic PCR-marker BvXY1 amplified a single specific 100–200 bp band in females of six green toad taxa, but 2–3 bands (5 in *B. turanensis*) in their males, one in females' size, presenting the X-copy. The additional male bands with the initial BvXY1-primers yielded identical or similar sequences indicating duplicates on the Y-homolog.

### Synteny and FISH-TSA of *bod1l* dismiss duplication with translocation

New sex determining genes evolve by allelic diversification or from gene duplications[42]. To elucidate the situation in *B. viridis*, we examined synteny and localization of the candidate locus. *Bod1l* lies between the genes *Cpeb2* and *nkx3-2* (NK3 homeobox 2), on the long arm of scaffold 1, an evolutionarily conserved gene order likewise found in *B. bufo*, *Rana temporaria* and *Xenopus tropicalis*, suggesting synteny throughout anurans.

Fluorescence in-situ hybridization with tyramide signal amplification (FISH-TSA) on metaphase chromosomes of a male *B. viridis* revealed two signals, one on each homolog, for a *bod1l*-specific probe at a position on the distal part of the short (p) arm of chromosome 1 (Fig. 2d). Two (and not three) signals suggest that this gene exists in two copies in XY-males of *B. viridis*, implying that both heterozygous sequences evolved by structural diversification[42] rather than translocation of a male-specific duplication, as shown for *dm-w* of female *Xenopus laevis*[19]. This result is further supported by the pool-seq and long-read male coverage (see below).

### Targeted sequencing in multiple related species sharing the sex marker

If the hypothetically sex-determining SNPs in coding *bod1l*-regions arose in a common ancestor, we expected the five diploid taxa, sharing BvXY1 in 5′-*bod1l*-regions (Fig. 2b), to share at least some exonic, possibly functional SNPs. To test this, we applied targeted enrichment to multiple females and males of five taxa, using an AmpliSeq custom DNA panel on 80 kb (93%) of the *bod1l*-X-copy, including its 5′-region with BvXY1, and analyzed 8183 SNPs. Of note, none of the *bod1l* coding SNPs, present in *B. viridis* (Supplementary Data 3), was conserved across all five toad taxa (Supplementary Data 4, 5, Supplementary Fig. 4a), suggesting that this coding region (cds) does not contain the mechanistic key to green toad sex determination. Instead, the targeted enrichment approach narrowed down the sex-specific region in the candidate locus to the 5′-region of *bod1l*, containing BvXY1 (Supplementary Fig. 4b).

### Y-chromosomal structure of *B. viridis* from long-read WGS

After first working with the female reference genome, X-haplotype and *bod1l*-coding region, we then moved the focus back to the 5′-end of *bod1l* and the BvXY1-vicinity on the Y-haplotype. To elucidate this Y-chromosomal structure, we reconstructed this region, which showed about 15x coverage (haploid), from ca. 30x whole genome long-reads of a single male *B. viridis* and aligned it to the X-copy of the *bod1l*-region. Female-to-male pool-seq-coverage of X- and Y-haplotypes yielded the expected diploid female and haploid male coverage (Supplementary Figs. 5, 6). Additionally, improved base-calling and haplotype-aware correction algorithms became available and enabled us to also assemble a whole male genome from that data, which supported the results of the manual approach (Supplementary Data 6).

In the *bod1l*-region, the confidently assembled Y-haplotype-specific sequence comprises about 55 kb with some greater-than-zero female pool-seq-coverage, explicable by repeat-enriched fragments, not assignable to X or Y (Supplementary Fig. 7). The male long-read whole genome assembly revealed several structural changes between X- and Y-haplotypes in the 5′-*bod1l*-region. The *B. viridis*-Y contains large insertions, mainly stemming from repetitive elements, and a non-coding RNA (ncRNA, see below). None of the so-called 'usual suspects', genes that often acquire sex-determination function in vertebrates[1], was detected in this region.

### Larval gene expression in sibling *B. viridis* tadpoles from multiple localities

Pool-sequencing of genomic DNA in *B. viridis* had revealed consistent heterozygosity in males for *bod1l*-exons 7 to 10. Accordingly, RNAseq revealed sex-differential expression (scf1:566,831,904-566,833,300), where heterozygous *B. viridis* males showed 1 SNP in exon 7, and 14 SNPs in exon 10, while females were always homozygous (confirmed by cDNA-based PCR-products). To test whether *bod1l*-products are differentially composed after transcription of X- and Y-copy, we analyzed splice sites in developmental transcriptomes, including the sex determination period. The absence of significant sex-specific differences excluded alternative splicing. However, this analysis allowed manual improvement of the *bod1l*-gene model.

To study the expression of the candidate sex determination gene (*bod1l*), the adjacent ncRNA-Y and ca. 80 other genes (Supplementary Data 7) playing roles in sexual development and/or sex determination in other vertebrates ('usual suspects'[1]), we then examined larval transcriptomes of six sibling groups, in total 54 individual toads from six locations, ranging from Gosner[43] stage 23 (day 10 after fertilization) until Gosner stage 46 (the last including some toads about 6 weeks after metamorphosis) as well as from two adult gonads.

Since gonadal differentiation is histologically detectable only at or after metamorphosis completion (>Gosner 46[25,35,43]), the critical period of genetic sex determination lies earlier, within our sampling period. In Gosner stage 46, 6 weeks after metamorphosis, we could anatomically distinguish between Bidder's organ and early testis, proofing sexual differentiation. Until Gosner stages 34–38 (Fig. 3), overall *bod1l*-expression varies in *B. viridis* between males and females. From Gosner-stages 38 to 43, other genes, specific for male sexual development, like *sox9*, *sox10*, *sox17*, and *sf1*, show higher expression in males than in females. Other such genes change their expression distinctly after Gosner stage 38, like *ck18* (bufVir1.18618.1) or *11βHD* (bufVir1.16559.1) or − in our series−were detected only in Gosner stage 46 and subadult testes (*dmrt1, amh*). Female-specific genes also changed their expression around Gosner stages 34–38; *wnt4*, *ihh*, *foxL2*, *ptch1*, *gli3*, *fshr* from Gosner stage 43 or 45; however, their expression was stronger in males than females. Expression of *gdf9*, *cyp19a1a*, and *cyp17a1* was detectable only in Gosner 46 and adult ovaries, with *cyp19a1a* and *cyp17a1* presenting good markers for the timing of gonadal sexual differentiation in anurans. *Bod1l*-expression differences between females and males (Fig. 3) lack statistical significance but occur at the right developmental period (Gosner stages 27–44), as required for a sex-determining gene. Early male transcriptomes (Gosner stages 23–44) contain also the bufonid-specific

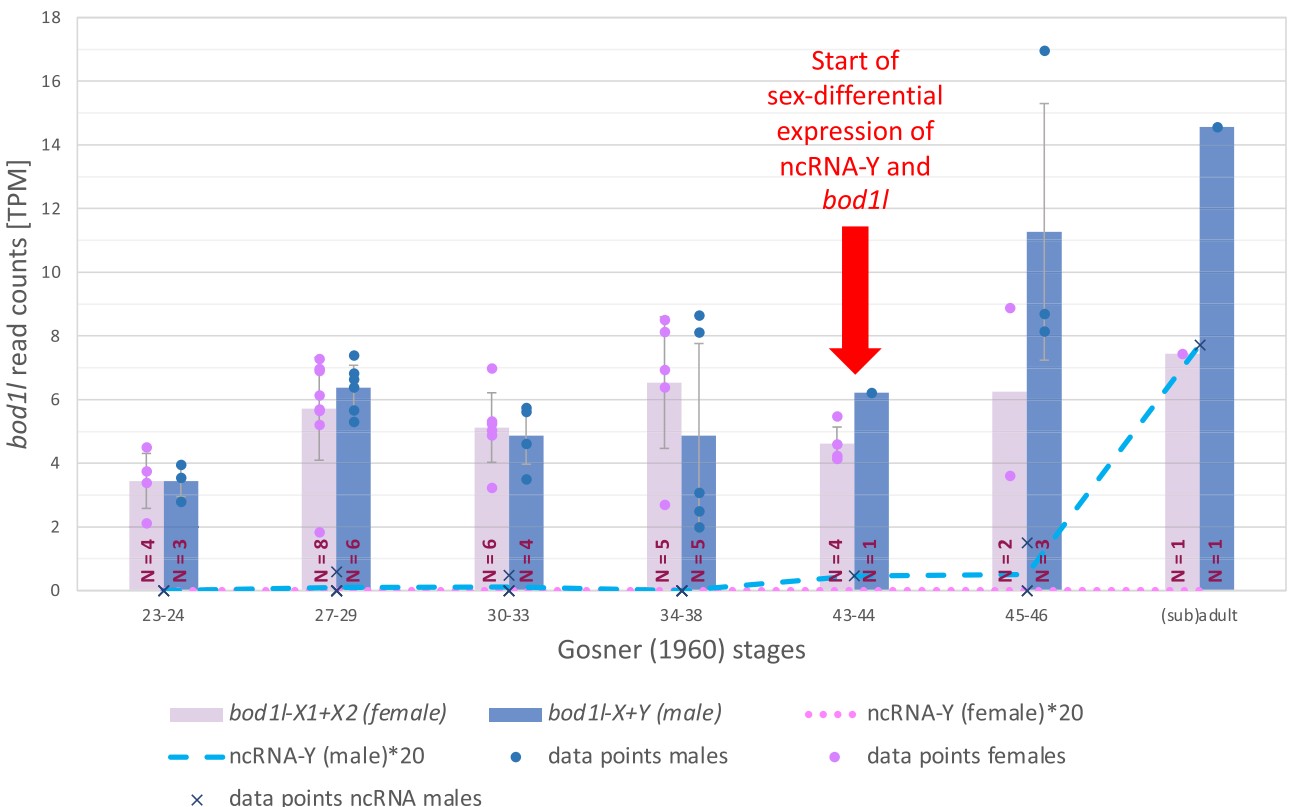

**Fig. 3 | Additive sex-specific XX- or XY-expression of *bod1l* and expression of the ncRNA-Y in females and males in seven developmental stages, with numbers (N) of females and males (N) per stage shown in bars, genetically sexed using PCR-marker BvXY1.** Gosner[43] stages 23–24, 10 days after fertilization, Gosner stages 27–29: 15 days after fertilization, Gosner stages 30–33: 18 days after fertilization, Gosner stages 34–38: 34 days after fertilization, Gosner stages 43–44: beginning of metamorphosis, Gosner stages 45–46, including toads up to six weeks after metamorphosis, the latter with anatomically visible differentiation of testes and Bidder's organs in males, and ovaries in females, subadult male and adult female. Numbers of available tadpole transcriptomes varied by stage; TPM: Transcript reads Per Million RNAseq reads; note that TPM for the ncRNA-Y were multiplied 20x. See Supplementary Fig. 8 for comparison of additive sex-specific expression of *bod1l* and ncRNA-Y and haplotype-specific *bod1l*-expression in males. Bars present mean values; error bars show the standard deviation of the respective data points.

Bidder's organ, considered similar to a rudimentary ovary, that develops adjacent to the anterior primordial gonads[25]. Published qPCR-data in other bufonids suggested that gene expression in adult Bidder's organs differs from that of testes and ovaries but resembles the latter, consistent with classical, mid-20th-century male-to-female sex-reversal experiments, when Bidder's organs grew and resumed oogenesis after the removal of testes[27] (reviewed[25]). In green toads, Bidder's organ develops in males but barely in females[25]. Given that the "ovary-like" primordial Bidder's organ is contributing tissue to our RNAseq-analysis of male tadpoles until Gosner stage 44, we assume a "female-bias" in males' transcriptomes (Gosner stages 23 to 44). In conclusion and in line with the AmpliSeq-data from multiple species, only sharing the BvXY1-marker across taxa, we consider the male-only expression of ncRNA-Y and the quantitative differences in additive *bod1l*-expression between males and females (Fig. 3) as sufficient to trigger green toad sex determination. Of note, there is a trend of greater Y-haplotypic expression of *bod1l* between Gosner-stages 38 and 43 (coinciding with an increase of the ncRNA-Y expression, next paragraph) and a distinctly greater additive *bod1l*-XY-expression in males than XX-expression in females (Fig. 3, Supplementary Fig. 8). *Bod1l*-Y also shows the highest expression in testis compared to other male organs (heart, brain, liver).

**A non-coding RNA (ncRNA) is Y-specifically expressed upstream of *bod1l***
When further comparing RNAseq (mapped to Y- or X-haplotypes) from ovaries *vs.* testes (in tadpoles plus adjacent Bidder's organ), we discovered a long non-coding RNA ("ncRNA-Y", ca. 1.2 kb, 4 exons, Fig. 4)

with its exons 1 and 2 being Y-specific, and expressed exclusively in males, as confirmed by PCR from cDNAs (Fig. 4c, Supplementary Fig. 9). The ncRNA-Y harbors the initially detected gDNA-marker (BvXY1) in intron 3. We hypothesize that this male-specifically expressed ncRNA-Y hints at an enhancer element, missing on the X, which regulates male *bod1l*-expression and simultaneously transcribes the ncRNA-Y in a one-dimensional (1D) manner from the opposite strand[44]. With ca. 80 kb, the putative enhancer sits in a reasonable distance to the *bod1l*-promoter (Fig. 4a, b)[45]. We also tested 200 bp-"sliding-windows" of the ncRNA-region for their in-silico potential as enhancers of *bod1l* (Supplementary Data 8; Supplementary Fig. 10). The most stringent threshold of 0.9 still predicted larger parts of the Y-specific region to function as strong enhancers, intensified by numerous G-quadruplexes (>25 alone in intron 1 of ncRNA-Y), known to accumulate at promoters and enhancers[46]. As shown by PCR using cDNA, this long ncRNA-Y is also expressed in skin, liver, brain and heart of males after metamorphosis as shown using cDNA, and exhibits simultaneous expression of the Y-haplotype of *bod1l* with the strongest signals in testis (Fig. 3). Due to its low expression level, it is only detectable in high-coverage transcriptomes of the primordial male gonads and adult testes but not in heart, brain and liver high-coverage transcriptomes. It comprises 5.8 kb on the gDNA level and can be male-specifically PCR-amplified in all those six taxa containing the BvXY1-marker and suggests to be a common ancestral feature (Fig. 4d).

**Spatially different gene expression in the developing gonads**
For a sex-determination gene, not only the accurate timing of its expression but also its organ- and cell-specific spatial expression is

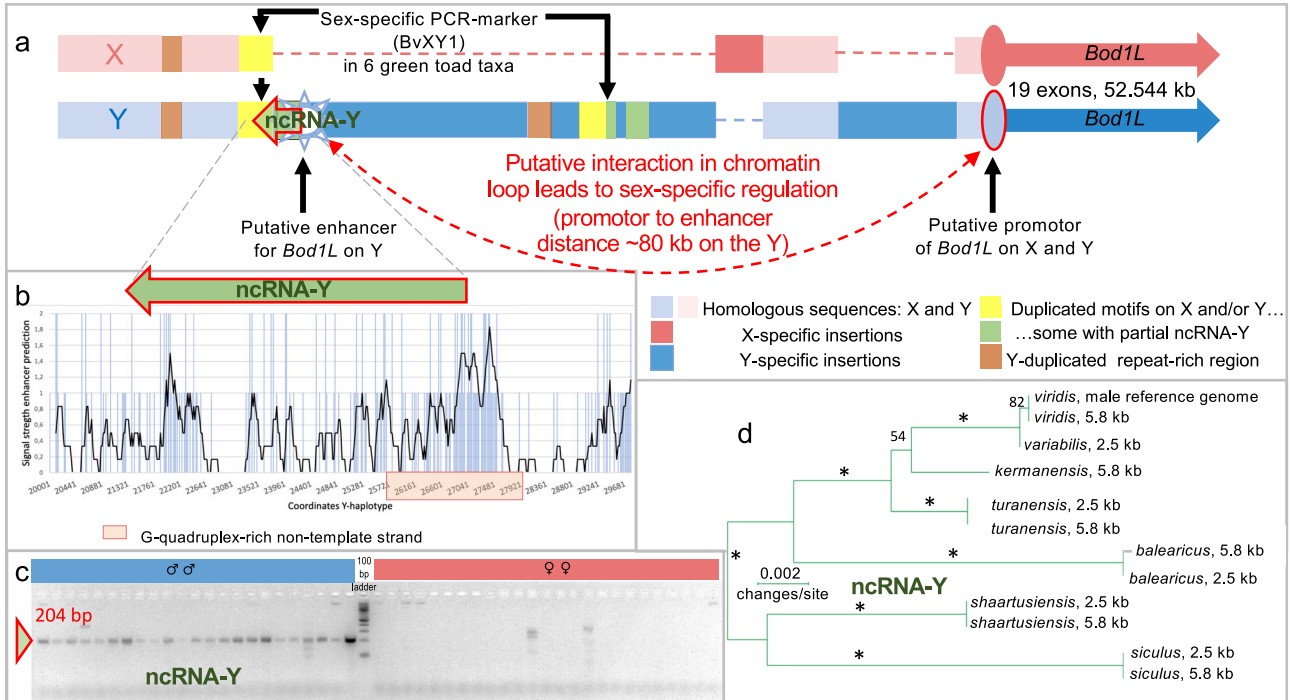

**Fig. 4 | Structure of the X- and Y-haplotype of the candidate sex locus of**
***Bufo(tes) viridis* with focus on the 5′-end of *bod1l* and the Y-specific ncRNA.**
**a** X-chromosomal (above, pink) and Y-chromosomal haplotype (below, blue) with
color-coded homologous or X- and Y-specific INDELs, and the Y-specific long (ca.
5.8 kb) non-coding RNA (ncRNA-Y), flanking a putative enhancer at the 5′-end of the
gene, ca. 80 kilobases before its promoter. **b** Graph of the signal of the AI-based
software *i-enhancer*, reaching from 0 to 2, predicting strong enhancer properties of
the Y-specific 5′-end of *bod1l*, including a G-quadruplex-rich region on the non-
template strand, overlapping with the ncRNA-Y. **c** Y- and thus male-specific
expression of the ncRNA-Y (204 bp PCR-product from cDNA) in larval green toads,

from 10 days after fertilization (Gosner stages 23–24) until early metamorphosis
(Gosner 46); in females, only non-specific products occurred, since X-copies of
*bod1l* do not contain this ncRNA (for details and positive controls of females:
Supplementary Fig. 9). **d** Unrooted Maximum-Likelihood-based phylogenetic tree,
showing that the ncRNA-Y evolved in a common ancestor of diploid Palearctic
green toads sharing this XY-system, with *siculus* and *viridis* spanning the most
divergent lineages; two PCR-products (2.5 and 5.8 kb), directly amplified by inde-
pendent PCRs from six related diploid green toad taxa, clustered by Y-gametologue
and thus species (* symbolized 100% bootstrap support; only support values > 50%
are shown).

essential. We provide evidence by RNA in-situ hybridization in histo-
logical sections of gonads in Gosner stages 36–37, where *bod1l* shows
sex-specific differential expression. In female gonads, *bod1l*-expres-
sion was evident in *vasa*-positive primordial germ cells (developing
oogonia) but absent from surrounding somatic pre-granulosa cells
(Fig. 5a, c, Supplementary Fig. 11). In contrast, *vasa*-positive male pri-
mordial germ cells (developing spermatogonia) as well as a majority of
adjacent somatic pre-Sertoli cells exhibited *bod1l*-expression
(Fig. 5b, d, green arrows). In Gosner stages 43–44, at the beginning of
metamorphosis, *bod1l*-expression in females remained similar to
Gosner 36–37. In males, however, *bod1l*-expression was no longer
detectable in somatic pre-Sertoli cells (Fig. 5e, f; Supplementary
Fig. 11). This suggests a dynamic regulation of *bod1l* in early male and
female gonads, with expression in somatic cells exclusively in male but
not female gonads. This precedes the differences observed by RNAseq
from trunk sections in Gosner stage 43–44, when quantitative differ-
ences of sex-specific *bod1l*-expression occur (Fig. 3).

It has been shown that *Bod1l* interacts with *Setd1a* to facilitate tri-
methylation of histone H3 at lysine 4 (H3K4me3[47]), with similarities in
eukaryotes from yeast to mammals[48]. To examine its status in male and
female gonads, we performed H3K4me3-immunostaining on Gosner
43–44 gonads (Fig. 5g, h). In females, gonads exhibited distinct
H3K4me3-signals in nuclei of both germ and somatic pre-granulosa
cells (Fig. 5g, white asterisks and arrowheads, respectively; Supple-
mentary Fig. 11). In contrast, H3K4me3 was strongly detected in male
germ cell nuclei (Fig. 5h, white asterisks) but hardly detectable or
absent in nuclei of somatic pre-Sertoli cells (Fig. 5h, yellow arrow-
heads). In Gosner stage 46, six weeks after metamorphosis, *bod1l* was

expressed (RNAseq, PCR of cDNA) in both sexes in several somatic
tissues (brain, heart, liver, intestine) as well as in gonads.

## XY-phylogeny in six Eurasian diploid taxa from targeted sequencing

For a sex-determining locus, evolved from a common ancestor, non-
recombining loci are expected to cluster by gametologue (i.e., X- or
Y-specific haplotypes) and not by species. Analyzing X- and Y-specific
sequences obtained from the AmpliSeq custom DNA panel and
amplicon sequencing of ncRNA-Y of six diploid green toad taxa, we
show that in the 2,671 bp-region (scf1:566,788,797-566,791,468), con-
taining the sex-specific BvXY1-marker, sequences cluster by gameto-
logue (Xs vs. Ys, Supplementary Fig. 4c), while in the similarly-sized
adjacent upstream region (scf1:566,786,116-566,788,796), sequences
clustered by taxon/species, suggesting occasional recombination
(Supplementary Fig. 4d). This provided evidence of both the stop of
recombination since the sequence arose in a common ancestor and
the apparent functional relevance of this region, since these species
also do not share coding sequences.

## Discussion

Our data on *B. viridis* and five related diploid green toad taxa are in
accordance with a locus that comprises a ncRNA and structural
changes in the 5′-regulatory region of a gene (*bod1l*), as well as sex-
specific ncRNA-expression in heterozygous (XY) males *vs.* homo-
zygous (XX) females, accompanied by strong phylogenetic and
poolseq-supported signal of local recombination arrest. Sex-specific
expression occurs at the right time, with male-restricted expression of

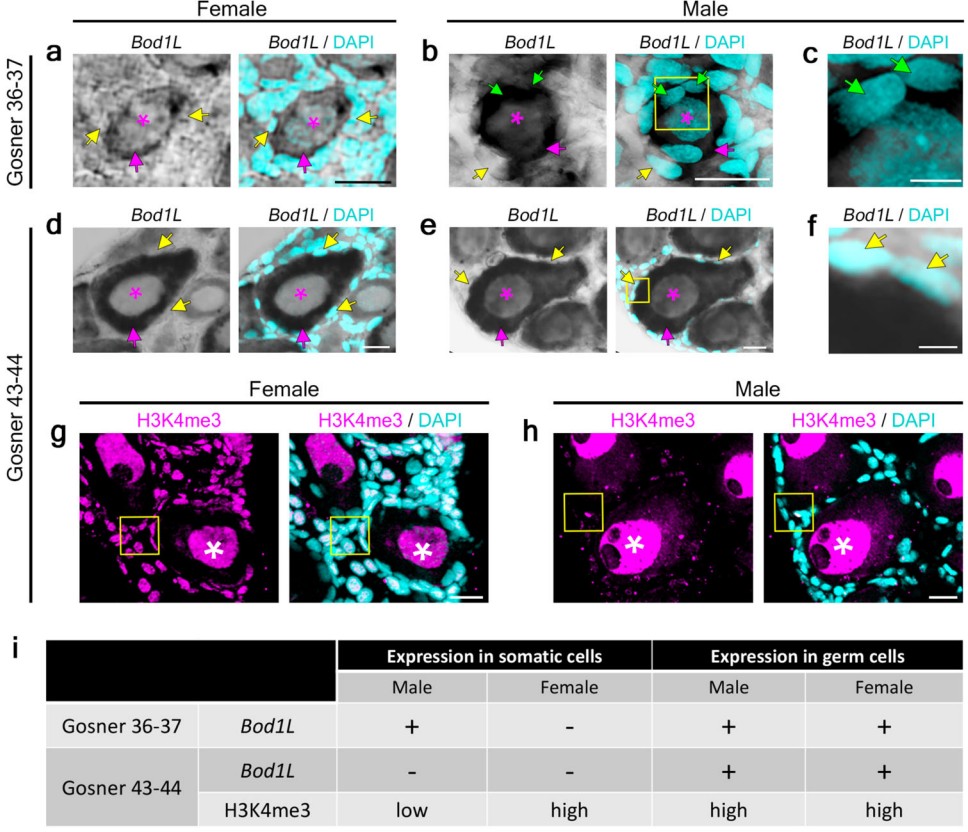

**Fig. 5 | *Bod1l*-expression and H3K4me3-methylation in primordial gonads of *B. viridis*, i.e., in the female early ovary, and in male Bidder's organ, contributing to the testes anlage (Gosner stages 36–37) as well as in early metamorphosis (Gosner stages 43–44).** *Bod1l* expression shown by RNA in-situ hybridization (ISH) on 20 μm transverse cryosections of female (**a**, **d**) and male (**b**, **c**, **e**, **f**, **h**) gonads at developmental stages Gosner 36–37 (**a**–**c**) and 43–44 (**d**–**f**), the latter presenting early metamorphic stages. DAPI was used to stain nuclei. Magenta asterisks indicate nuclei of germ cells; magenta arrows: cytoplasm of germ cells; yellow arrows: nuclei of somatic cells not stained by *bod1l* RNA ISH; green arrows: nuclei of somatic cells

stained by *bod1l* RNA ISH. Yellow boxes in **b** and **e** indicate zoomed images shown in **c** and **f**, respectively. H3K4me3 and DAPI staining on 20 μm transverse cryosections of female (**g**) and male (**h**) green toad Bidder's organ and gonads, Gosner stage 43–44. White asterisks indicate H3K4me3-positive germ cell nuclei; yellow boxes indicate somatic cells. Scale bars (**a**, **b**, **d**, **e**, **g**, **h**): 20 μm; (**c**, **f**) 5 μm. **i** Summary of *bod1l*-expression and H3K4me3-methylation in somatic cells and germ cells in Gosner stages 36–37 and 43–44 male and female gonads. For each Gosner stage, five sections from one gonad sample were analyzed in two independent experiments.

the ncRNA from the Y, much earlier than histologically detectable differentiation, and *bod1l*-expression in the target organs (primordial gonads), and therein in the germ and surrounding somatic cell populations (in male pre-Sertoli but not in female pre-granulosa cells). These properties suggest that the region containing ncRNA-Y and *bod1l* presents a candidate sex determination locus. Future evidence (e.g., using CRISPR/Cas9) is desirable for its better mechanistic understanding. A single genome-wide peak of heterozygosity between female and male revealed by pool-seq and coverage differences on whole genome assemblies of both sexes (Fig. 2a), along with evolutionary conservation of the structural XY-differentiation in several Eurasian taxa sharing the Y-specific ncRNA, support its candidacy as the Palearctic green toad sex determining locus. The expression of the ncRNA-Y and *bod1l* in Bidder's organ of the differentiating male gonad (Gosner stage 46) may contradict the view that Bidder's organ in this XX/XY system can be simply interpreted as a 'proto-ovary'.

The only known amphibian master sex determining gene, *dm-w* of clawed frogs (*Xenopus*), governs female-heterogamety (ZW♀/ZZ♂) and evolved by gene duplication as a *dmrt1*-paralog, involving neofunctionalization of a non-coding part of a DNA-transposon[12,19–22,49]. Here, we provide evidence for a sex-locus in a male heterogametic bufonid toad system (XX♀/XY♂), evolved by structural diversification in a common ancestor, as shown by the associated ncRNA-Y-phylogeny (Fig. 4c). To our knowledge, this reveals only the 2nd master sex determining locus in amphibians.

Our data suggest either a sex-determining function of the ncRNA-Y itself or its functional interaction with *bod1l*, e.g., as an enhancer[50,51].

Long non-coding RNAs (lncRNAs) play important roles in gonadal development and sex differentiation[50,52], and thus the ncRNA-Y might suffice as a master sex determinator, although, so far, we have not detected homology with genes known to play roles in sexual development. We rather consider a regulatory effect on the neighboring *bod1l*. The ncRNA-Y, harboring the partially duplicated BvXY1, contains a neighboring signal for a G-quadruples-rich enhancer towards its 3'-end. This is a mechanistic hint that the enhancer modifies *bod1l*-expression in XY-males (Fig. 4), as shown by timely and spatially differential expression in germ and pre-Sertoli but not in pre-granulosa cells (Fig. 5).

There are also several, partly interlinked pathways by which spatially and/or timely differential *bod1l*-expression may determine sex in green toads.

(i) Sex determination of vertebrates differs regarding the timing of primordial germ cells to enter meiosis[53], with females entering meiosis earlier than males, exhibiting an arrest. *Bod1l* is well-known to maintain genomic integrity at sites of DNA-replication damage[54]. *Bod1l* interacts towards its N-terminus with the histone methyltransferase *Setd1a*, which triggers the histone chaperone function of *Fancd2*[55], part of the Fanconi anemia group (*Fanc*), that plays roles in zebrafish sex reversal[56]. Thus, differential *bod1l*-expression might cause a compromised DNA repair system by shifting sex determination towards males.

(ii) Interaction of *Bod1l* with *Setd1a* and H3K4me3 refers to tri-methylation on lysine residue 4 in the amino-terminal tail of histone H3, is associated with potentially active promoter regions[57] and linked to double-strand breaks, essential for meiosis 1 crossing-over, ensuring chromosome segregation[58,59]. Thus, the differential *bod1l*-expression might enable double-strand breaks as a precondition for the earlier start of recombination in females but not in males.

(iii) H3K4-methylation in mammals[57] is associated with de novo DNA-methylation in germ cells, and essential for maintaining transcription in the growing oocyte and for triggering zygotic genome activation[60], with XY-mice deficient for the H3K9-demethylating enzyme *Jmjd1a* showing increased H3K9-dimethylation and decreased activating marks of H3K4-trimethylation across the *Sry*-locus, leading to a high frequency of sex reversal[2].

According to Dupont and Capel[61] "prior to sex determination" in placental mammals, bipotential supporting gonadal cells of both genetic sexes maintain identical transcriptional programs, allowing commitment to either Sertoli (testes) or granulosa (ovary) cell fate. In pre-Sertoli and pre-granulosa cells, bivalent promoters of key sex determining genes (e.g., *Sox9*, *Fgf9*, *Wnt4*, *Foxl2*) have both, H3K27me3 and H3K4me3 marks[61]. After sex determination by the master sex determination gene *Sry*, promoters to be activated in that sex lost the repressive H3K27me3 mark, while genes to be silenced maintained their bivalency[62]. This suggested that (a) sex-determining genes are poised for activation (or repression) at the bipotential stage, (b) gene activation involves removal of H3K27me3 marks, while (c) repression of the alternative supporting cell fate requires maintenance of existing H3K27me3 repressive marks rather than de novo deposition of these marks after commitment to the Sertoli or granulosa program[61].

One hypothesis is that differential *bod1l*-expression (reminiscent of *Sry*) might determine sex by changing H3K4-methylation in green toads during early gonadal development, prior to metamorphosis.

To test the hypothesis that *bod1l-expression* coincides in time and space with changes of H3K4-methylation in developing green toad gonads, we used H3K4me-antibodies together with RNA in-situ hybridization (Fig. 5). In Gosner stage 43–44, slightly prior to morphological differentiation, we found a strong difference between male and female gonadal somatic cell H3K4-methylation. This suggests that spatially different *bod1l*-expression results in sex-specific methylation patterns that initiate sex-specific gonadal differentiation. However, opposite to the situation in mammals, H3K4me3 is weaker in somatic cells of green toad males than those of females (Fig. 5), while increasing H3K4-methylation allows *Sry*-expression activation (male determination). Another major difference is that the primordial gonad of bufonid toads, specifically in green toad males, includes the "ovary-like" Bidder's organ.

Given the multiple potential links between *bod1l*-expression and sex-specific meiotic stages shown above, we hypothesize that this gene may function as a suppressor of the female sex determination pathway – a scenario found in many vertebrates[63]. Following differential *bod1l*-expression, in the bipotential gonad, ovarian pre-granulosa cells transdifferentiate to testicular pre-Sertoli cells.

In contrast to the only known amphibian master sex determining gene in clawed frogs (*Xenopus*), the green toad sex-locus with ncRNA-Y and *bod1l* did not evolve by gene duplication but structural diversification in the 5′-region of the gene, namely by Y-specific sequence insertion. This evolution of sex determination seems broadly similar to allelic diversification, so far unknown in amphibians, but known from many examples of master genes in fish[42].

## Methods

### Ethics and Animal Welfare Statement

Our research complies with all relevant ethical regulations; the relevant animal protection committee of the IGB and the city of Berlin (LaGeSo) approved the study protocol where relevant. Most adult toads were sampled more than 20 years ago by either minimal invasive sampling methods using tail tips of larvae or saliva of adults, obtained by buccal cotton swabs. We also used tissues of road kill toads and/or collected a few adult animals in Greece. For sampling and export of toads and tadpoles, permits were kindly provided by the Ministry of the Environment of Greece (115790/229) 2014–2017. Developmental stages of tadpoles were obtained from control groups of an animal experiment approved by the German State Office of Health and Social Affairs (LaGeSo, Berlin, Germany; G0359/12) and a commercial breeder. Developmental stages of larvae (tadpoles) and early metamorphosed toads were observed under a binocular microscope and determined using the drawings of developmental stages (table) by Gosner[43]. Tadpoles were reared in a 12/12 h light/dark cycle at constantly 22 ± 1 °C in sufficiently aerated and regularly cleaned tanks. Weekly monitored water parameters comprised: dissolved oxygen, nitrate, ammonium, pH, conductivity, and hardness and kept as described[64]. Tadpoles were fed SeraMicron (Sera, Germany), later supplied with TetraMin (Tetra, Germany). To imitate natural conditions at metamorphosis, animals were transferred to glass terraria at Gosner stage 46. Thus, the tadpoles and toads were kept according to approved practice (permit ZH 114 to IGB), including the provision of appropriate tank size, sufficient rate of waterflow, natural photoperiod, *ad libitum* food supply, as well as temperatures within the species' thermal tolerance range. This ensured that no pain, suffering, distress or lasting harm was inflicted on the animals. According to the permit G0359/12, tadpoles and juvenile (and two adult) toads were euthanized by an overdose of Tricaine PHARMAQ 1000 MG/G (MS222) (concentration: 500 mg/L) since this does not incur pain, suffering or distress.

### Chromosome-scale female genome assembly

The female *B. viridis* reference genome was sequenced in-house at IGB using ONT long reads (MinIon device, SQK-LSK109 kits and R9.4.1 flow cells) and Illumina short reads (Novogene, Europe). The long reads and pre-assembled short reads IDBA-UD assembly[65] were assembled using the WTDBG2.2 assembler[66]. Sequences were error-corrected using two rounds of Flye polish with long reads[67] and four rounds of VCFconsensus using short reads[68]. The chromosome-scale assembly was achieved using Arima HiC-data and the JUICER/3DDNA package and manual curation by juicebox[69–71]. De novo repeat analysis of the assembly was performed by the RepeatModeler/RepeatMasker package. Protein-coding genes were annotated using RNAseq data of *Bufonidae* as well as Hyloidea transcriptomes and more distantly related amphibian proteins, available at NCBI (Supplementary Text 1). A reference genome browser was created http://genomes.igb-berlin.de/Greentoad/[72].

### Assembly of Y-haplotype

Several iterative approaches were used to extract Y-haplotypic sequencing reads of *B. viridis* GR57 male[73] long read data (Oxford Nanopore Promethion platform; Novogene). First, we mapped the reads to the BvXY1-marker sequences to separate Y reads of the region. Second, we performed a FLYE[67] assembly of the male data, identified several contigs with variation compared to the X haplotype (*bod1l*-region), and extracted the corresponding reads. The Y-specific reads were assembled, then added to the female assembly and all male reads were mapped to the combined sequences, which allowed finding more reads matching better to Y than to X contigs. This procedure was iteratively repeated several times until the Y-haplotype and its structure could be assembled. Pool-seq data was again mapped to the female assembly with added Y-haplotype sequences by MINIMAP2[74] showing the expected coverage differences between female and male pools. A dot plot comparison of X versus Y haplotype involved the corresponding function of the NCBI Blast website.

## Assembly of a male genome

Data that was used for the manual assembly of the Y-haplotype (see above), served also to perform a complete de novo assembly of a male *B. viridis* individual. To improve assembly results RAW data was first base-called by the most up-to-date basecaller for ONT data (dorado-0.5.1-linux-x64 with dna_r9.4.1_e8_sup@v3.6 model). Then the reads were corrected using the haplotype-aware read correction method, implemented in PECAT[75]. The corrected reads and the prior flye assembly contigs (see above) were assembled using wtdbg2. Contigs were corrected by one round of Flye polish (with corrected long-reads) and two rounds of VCFconsensus (with male poolseq data). Male Hi-C data was mapped to the contigs using CHROMAP[76] and chromosomal scaffolds were computed by YAHS[77]. Some unplaced contigs could be placed into the YAHS scaffolds using RAGOUT2[78] and synteny analysis with the female genome. The assembly was scored using BUSCO analysis and had highly similar scores as the female assembly.

## Double-digest RAD sequencing (ddRADseq)

DNA of three genetic families consisting of one data set with both parents (buccal swabs) and 53 offspring, of which 20 females and 17 males were phenotypically sexed using gonad histology, while in the remaining 7 females and 8 males, genetic sex was pre-assigned using sex-linked microsatellites. The two other sample sets comprised 24 (14 females, 6 males, 4 unsexed individuals), or 15 siblings (3 females, 3 males and 9 unsexed individuals) without known parents. DNA was extracted using the DNeasy Tissue Kit or the BioSprint robotic workstation (QIAGEN), following the manufacturer's protocols. Double-digest RADseq libraries were prepared according to this study[79]. In brief, genomic DNA digestion by two restriction enzymes (*Sbf*I, *Mse*I) and subsequent ligation of fragments to adapters containing unique barcodes (4–8 bp) was followed by library-PCR for 20 cycles for only different cut-sites containing fragments, and a gel-based size selection, isolating the 400–500 bp fraction. RADseq libraries were single-end sequenced on an Illumina HiSeq 2500, at a 20x coverage.

Raw ddRAD reads were demultiplexed with STACKS *process_radtags* version 2.55[80], using the enzymes *Sbf*I and *Mse*I (*--renz_1 sbfI --renz_2 mseI*) and the parameters -c (clean data), -r (rescue barcodes and RAD-Tag cut sites), and -q (discard reads with low quality scores).

Demultiplexed reads were used as input for the RADSEX workflow[81]. First, a table containing the number of occurrences of each unique sequence in the reads file in each individual was generated with *radsex process*, using default parameter values. Then, the distribution of markers between males and females was computed with *radsex distrib* using a minimum depth to consider a marker present in an individual of 5 (--min-depth 5). Markers were aligned to the genome of *B. viridis* (bufVir1) using *radsex map* using a minimum depth of 5 (--min-depth 5) and other parameters set to default. Results were plotted with the *SGTR* package, with the entire process provided as a Snakemake workflow[82].

## Poolseq approach using phenotypically sexed toads: 'poolsex'

Of 22 safely sexed adult toads of each sex from multiple populations in the range of the European green toad taxa (*B. viridis viridis* and *B. viridis variabilis*), using buccal swabs or tissues genomic DNA was extracted using the DNeasy Tissue Kit or the BioSprint robotic workstation (QIAGEN), following the manufacturer's protocols. Individual DNAs were quantified using the QuantiFluor® dsDNA System kit (Promega), adjusted and equimolarly mixed to yield a female and a male pool. Whole genome library preparation for 300 bp fragments and whole genome sequencing was performed at NOVOGENE (UK), on a NovaSeq PE150 to yield 131 and 140 Gbp of data for male and female pool, respectively. Reads were mapped to the reference genome using the short-read mode of MINIMAP2[74] (parameters -K 2000M -I 100 G -a

-x sr -t 80, and converted to sorted bam-files by SAMTOOLS[83]. Variants (SNPs, MNPs, INDELs) were called from both bam files using the PLA-TYPUS variant caller[84] with integrated duplicate fragment reads removal (−filterDuplicates = 1) and reassembly of reads (−assembleAll = 1). The resulting vcf-file was screened for XY- or XX-specific variants using AWK-scripting and different variant read coverage cutoffs (i.e., SNP read coverage in females and males >=15; variant read count in females <=7% or >=93% of total SNP read coverage as a condition for homozygosity; variant frequency in males between 45–55% of total SNP read cov. as a condition for heterozygosity; for screening ZW SNPs conditions for males and females were interchanged). Variants were clustered according to their distance in the genome (maximal distance of variants in a cluster: 9000 bp), counted using BEDTOOLS (merge, annotate[85]) and sorted by variant count. Regions with highest variant counts or densities and neighboring regions were inspected for genes related to sex determination. The same heterozygous SNP analysis was repeated using a male genome assembly (see above). Additionally, an analysis of male-specific pool-sex coverage of the male genome was performed. Here, we used the bam files (MQ60 reads only) to analyze bp-level sequencing coverage using BEDTOOLS genomecov. The resulting bedgraph files for female poolseq and male poolsex were combined by BEDTOOLS unionbedg. Coverage of both poolsex datasets was normalized by their modal coverage peak (setting diploid coverage => 1.0). Then bp positions were extracted, if they had values of 0.25 - 0.75 for male- and <0.05 for female-poolsex coverage. These genomic positions were clustered by distance and the number of male-specific covered bases in each cluster was counted BEDTOOLS(merge, annotate).

## Sex-diagnostic PCR marker design

The region 566,783,058-566,848,882 bp on scaffold 1 includes several INDELs apparent between the X and the Y haplotype, allowing design of a number of primers that were primarily tested in gradient-PCRs on phenotypically sexed *B. viridis*. While several primers showed sex-specific amplification in this species, one (BvXY1) could be optimized to work across multiple species.

A collection of DNAs (extraction: see above) comprising samples of phenotypically sexed green toads of the entire radiation[36] served to test sex-specificity of all PCR markers. PCR-amplifications were carried out in 25 µL reactions containing 1.25 µL of each primer (BvXY1_F: 5′_CTGAATAGCAATTGGACACAGC_3; BvXY1_R: 5′_ATGTGA-GAGCTGTTACAATGGA_3′), 14.5 µL of water, 5 µL of AllTaq buffer (5x), 0.5 µL of AllTaq polymerase (Qiagen), and 2 µL of DNA (concentration 4–20 ng/µL). Temperature regime comprised 2 min at 95 °C (initial denaturation), and 38 cycles of (1 min at 95 °C denaturation, 45 s at 59 °C annealing, 45 s at 72 °C extension), and 5 min at 72 °C final extension. PCR products (4 µL) were visualized by electrophoresis on 4% high-resolution Sieve-agarose (Biozym) gels, run for 60 min at 70 V.

To PCR-amplify either a 2.5 kb fragment or nearly the entire ncRNA-Y, comprising 5.8 kb based on gDNA on the Y-chromosome in all diploid green toad taxa with sex-specific BvXY1-markers, we used the same ingredients and volumes as above either with primers ncRNA_Y_ex1F: 5′_GGGCAGTGAAGAGAAGTCCT_3′, ncRNA_Y_ex2R 5′_TTCACCATTCTCCAGCCTGT_3′ for the 2.5 kb-fragment or 5.8kb_ncRNA_Y_F: 5′_GTGGTAGTTGCTGTGCCTTT_3′, 5.8kb_ncRNA_Y_R: 5′_CCGGTGTGGTGTAAAATATCCA_3′ for the 5.8 kb-piece. Temperature regimes involved 2 min at 95 °C, and 37 cycles of (1 min at 95 °C denaturation; 2.5 min at 63 °C (for the 2.5 kb) or 5 min at 59.4 °C (for 5.8 kb) annealing; 1 min at 72 °C extension and 5 min at 72 °C final extension (note extended annealing times). PCR products (4 µL) were visualized by electrophoresis on 1.5% agarose (Biozym) gels, run for 60 min at 80 V. PCR-products (21 µL) were barcoded using the NBD196 and LSK110 ONT-kits, and sequenced using the ONT-MinIon device. Basecalling was performed using ONTs GUPPY version 6 and the super accuracy base-calling model (SUP-model).

## Phylogenetic tree of a 2.7 kb-region at the Y-specific region boundary

Consensus sequences of 100 ONT reads of Y-specific amplicons were calculated using NGSPECIESID[86]. Homologous X-sequences were reconstructed from targeted enrichment sequencing data of females using BCFTOOLS consensus[87] for the region scf1:566,788,797-566,791,468. We also excluded regions of the sequence not covered by enrichment data by supplying a masking file through parameter "-m" of bcftools consensus. All resulting sequences were multiple aligned using MAFFT (automatic alignment parameters). The alignment was filtered by GBLOCKS[88] to remove columns with excessive missing data. IQTREE2[89] was used with parameter "-alrt 1000 -bb 1000" to calculate a phylogenetic tree. To calculate a phylogenetic tree of the upstream neighboring region (scf1:566,786,116-566,788,796) data from target enrichment for males and females was used as described above.

## Phenotypic sexing in multiple species

Phenotypic sex of adult toads of multiple diploid Eurasian green toad taxa had been established during the breeding period when males exhibited secondary sexual characters like nuptial excrescences and thorny glandular tips on the back, or were caught producing male advertisement calls and/or male release calls. Females were identified as filled with eggs, caught in amplexus, and/or while spawning, producing female release calls and/or lacking the male secondary sexual characters. During the spawning migrations, several road kill toads also allowed for anatomical identification (possession of ovaries or testes) of phenotypic sex.

## Testing enhancer activity of the Y-haplotype

We used the AI-based predictor "iEnhancer-Deep" in a 200 bp "sliding-window" approach in the ncRNA region to test its in-silico potential as enhancers of *bod1l*-expression[90].

## Developmental gene expression in sibling tadpoles from multiple locations

In total, 54 transcriptomes of sibling tadpoles comprising brothers and sisters (plus three unsexed larvae) of four genetic families from four different geographic locations as well as gonadal tissue of a subadult male and adult female from two additional localities were examined by RNAseq (transcriptomics). Tadpoles presented five developmental stages from untreated control groups of a previous experiment[91]. They were euthanized in MS222 and flash-frozen in liquid nitrogen. In tadpoles older than Gosner stage 37, the intestine was removed prior to flash-freezing. In tadpoles at Gosner stage 43–44, only the body segment that contained the gonads was used for RNA extraction. In toadlets of Gosner stage 46 (specifically 6 weeks after metamorphosis), ovaries, testes (separated from Bidder's organ), and several other organs (brain, heart, liver, intestine) were dissected and stored in RNAlater prior to flash-freezing. DNA from tailtips, or skin samples of each tadpole or buccal swabs (juvenile toads) served for genetic sexing using BvXY1-primers. RNAs were extracted using the TRIzol Reagent (Thermo Fisher Scientific, Waltham, USA) according to the supplier's recommendation and cleaned using the RNeasy Mini Kit (Qiagen). All RNAs included in this study exhibited RIN values > 9.0. Library processing and RNAseq were carried out by NOVOGENE (Cambridge, UK) on a NovaSeq 6000 PE 150, generating 9 Gb of raw data per sample.

Transcriptome sequences of 54 male and female juvenile developmental stages and two adult gonads were normalized and mapped to the genome using the nf-core RNA-Seq pipeline in STAR aligner (https://github.com/alexdobin/STAR/releases,–quantMode Gene-Counts). A list comprising genes that play roles in sexual development and/or sex determination in other vertebrates (also known as 'usual suspects'[1]) is shown as Supplementary Data 7. To plot *bod1l* and *ncRNA-Y* expression trends over the developmental stages and separate expression analysis of *bod1l-X* and *bod1l-Y*, we mapped all reads,

including the male haplotype, to the female assembly. As the *bod1l-Y* gene model is 3'-partial, we used the corresponding parts of the *bod1l-X* gene model for quantification. Counts of reads matching the different haplotypes' gene models were extracted using FEATURECOUNTS (from SUBREAD-2.0.3 package[92] with mapping quality (MQ) higher than 19, to increase haplotype specificity. Read counts were normalized by total mapped reads of each sample (TPM normalization). Average expression and standard deviation were calculated and plotted combining samples of the different stages using Microsoft Excel software.

## cDNA of developmental stages

Using the iScript RT Supermix for RT-qPCR kit (Biorad), 10 μL of RNA (concentration of 80 ng/μL) from all developmental stages (as used for RNAseq) were randomly reverse-transcribed into cDNA according to the manufacturer's recommendations. To test allele-specific *bod1l*-gene expression, we designed primers (bod1l_RNA1F: 5'_AACTGGAA-GAAAAGCGCAAA_3'; bod1l_RNA1R: 5'_GTGCTCTTGGCATTCTGTGA_3') and amplified 577 bp from cDNA under the same conditions as for BvXY1 that were Sanger-sequenced in both directions. To test expression of the ncRNA-Y, we used primers designed to match its inter-exonic regions NcRNAY_Ex1to2_F: 5'_CATGAGGAGAGCAGTAATGAACC_3', NcRNAY_Ex3to2_R: 5'_GCTGTCATCAACCTGCTCAA_3' and amplified and visualized 204 bp male-specific products from cDNA under the same conditions as for BvXY1 but at 55 °C annealing temperature.

## Probe preparation for RNA in-situ hybridization

The sequence marker obtained by cDNA amplification using bod1l_RNA1F/R primers was also used as a probe for RNA in-situ hybridization on histological gonad sections of various developmental stages. As a probe of a control gene, expressed in gonads of both sexes, we used a 501 bp cDNA fragment of *vasa* (DDX4-DEAD-Box-Helicase-4, VASA-homolog), amplified with the primers Vasa1F: 5'_AACCAAAAA-GAAGGCGGATT_3'; Vasa1R: 5'_ATGCCAGGATTGCTGTAACC_3' under the same conditions as for BvXY1 but with annealing temperature of 57 °C.

## RNA in-situ hybridization and H3K4me antibody staining

We opened the body cavity of freshly euthanized tadpoles of three developmental stages and fixed them in 4% paraformaldehyde (PFA) buffered in 1x PBS (Thermofisher), pH = 7.0, rotating at 4 °C overnight. Samples were washed in filtered 1× phosphate-buffered saline with tween 20 (PBST, ROTI®Stock) twice for 5 min, then dehydrated in a series of methanol (MetOH) with shaking at room temperature (25% MetOH in PBST for 10 min to 1 h, 50% MetOH in PBST for 10 min to 1 h, 75% MetOH in PBST for 10 min to 1 h, 100% MetOH for 10 min to 1 h, timing dependent on tissue sample size). Samples were stored in 100% MetOH at −20 °C until histological sectioning.

Cryosectioning, RNA in-situ hybridization and immunohistochemistry were performed as described[93] with slight modifications. Transverse cryosections (20 μm) were collected on SuperFrost Plus glass slides (Menzel Gläser J1800AMNZ) and stored at −20 °C for up to 2 months. To generate *bod1l* and *vasa* riboprobes, primers Bod1_RNA_Bv_1F (5'_AACTGGAAGAAAAGCGCAAA_3') and Bod1_RNA_Bv_1R (5'_GTGCTCTTGGCATTCTGTGA_3') were used to amplify a 577 bp *bod1l* sequence and primers vasa_1F (5'_AACCAAAAA-GAAGGCGGATT_3') and vasa_1R (5'_ATGCCAGGATTGCTGTAACC_3') were used to amplify a 501 bp *Vasa* sequence from Gosner 36–37 *B. viridis* cDNA. These sequences were then ligated to a pCR-bluntII-TOPO vector (Invitrogen) and plasmids were purified using a plasmid DNA miniprep kit according to manufacturer's instructions (Promega A1460). For synthesis of sense and antisense riboprobes, plasmids were linearized using *BamHI* or *NotI* (New England Biolabs) and in vitro-transcribed using T7 or SP6 polymerase (DIG RNA Labeling Mix,

Roche), respectively. Rabbit polyclonal anti-histone H3 (tri methyl K4) primary antibody (1:100, Abcam ab8580) and secondary goat anti-rabbit Alexa Fluor 568 (1:500, Invitrogen A-11011) were used. Sections were counterstained with 0.25 μg/mL DAPI and mounted in Mowiol 4−88 (Calbiochem). Imaging was performed using the Nikon SMZ18 stereomicroscope and Olympus FluoView FV3000 confocal microscope. To process images, Bitplane Imaris and Adobe Photoshop were used.

### FISH-TSA
As a probe for fluorescence in-situ hybridization with tyramide signal amplification (FISH-TSA), we designed novel primers to amplify a longer portion of *bod1l* from cDNA (*bod1l*_Exon7To10_F: 5'_AACTG-GAAGAAAAGCGCAAA_3', *bod1l*_Exon7To10_R: 5' _TGTGTTGTCTGGAATCACTGG_3'; same conditions as for BvXY1 but with annealing Temperature of 60 °C.), exhibiting the highest heterozygosity in the coding segment of the gene in *B. viridis* males. A PCR-product was produced using the PPP Master Mix (Top-Bio, Czech Republic) according to manufacturer's instructions and purified[94]. The cDNA amplicons were cloned using the TOPO-TA cloning kit and DH5α-T1R Competent Cells (Thermo Fisher Scientific). Plasmid DNA was isolated from bacterial colonies using the EZNA Plasmid DNA Mini Kit I (Omega Bio-Tek Inc.), sequenced, and tested to contain the correct insert by BLAT search on the green toad genome browser (see Data availability). The sequence of the *bod1l* clone was compared to the reference genome. Subsequently, the *bod1l* locus was re-amplified and purified[94]. A 20-year-old tissue suspension (bone marrow), obtained from a male European green toad from Germany, treated with colchicine and fixed in methanol-acetic acid[31] was dropped onto a microscope slide[95], followed by FISH. The probe was labeled with digoxigenin-11-dUTP (Jena Bioscience) using DecaLabel DNA Labeling Kit (Thermo Fisher Scientific). The FISH-TSA followed[96] with minor modifications[97]. Mitotic metaphase spreads were counterstained with ProLongTM Diamond Antifade Mountant with the fluorescent 4',6-diamidino-2-phenylindole, DAPI stain (ThermoFisher Scientific). Microscopy using Leica Microsystem (Wetzlar, Germany) and processing of metaphase images followed[98].

### Analysis of potential alternative splicing of *bod1l*
For the entire *bod1l* region, including the 3'- and 5'-UTR-regions (scf1:566495375-567417162), aligned RNAseq-reads were extracted from each RNA-seq alignment file, the splice site locations were retrieved and counted. Using all the splice sites, a reference file of them was built. The splice event counts were merged for all samples using the reference and stored as csv text file (Supplementary Data 9).

### Structural evaluation of the *bod1l*-X and Y protein copies
We used Alphafold2[99] via its cloud-based application and its local installation ColabFold[99] and analyzed the structure predictions by RaptorX[100] for structural and potential functional amino-acid differences between the X and the Y-copy of *B. viridis*. The molecular data viewer iCn3D[101] was used to inspect the structures and highlight specific amino acid residues (Supplementary Text 1).

### AmpliSeq custom DNA panel
After providing pre-publication access to the *B. viridis* genome-browser, the Illumina concierge design service team developed two primer pools covering ca. 80 kb of the *bod1l*-region (scf1:566,785,000-566,865,000) and 10 kb for a control gene (*amh*), known to play roles in sexual development and sex determination in other vertebrates and lying on the same scaffold (scf1:324181812-324192250). Overall, the theoretical coverage was 92.3% for *bod1l*, and 98.2% for *Amh*.

Wet lab approaches followed the "AmpliSeq for Illumina On-Demand, Custom, and Community Panels" (Document #1000000036408 v09; pp. 5−23 https://support.illumina.com/ downloads/ampliseq-for-illumina-custom-and-community-panels-reference-guide-1000000036408.html) for two primer pools with 193−384 primer pairs in the standard workflow. Briefly, this included ten steps, (i) quantification of DNA samples of multiple sexed green toads (QuantiFluor® dsDNA System kit, Promega), (ii) target amplification in two separate wells for the two primer pools, (iii) partial digestion of amplicons, (iv) ligation of individual indices/sample, (v) library clean up and (vi) amplification, (vii) 2nd clean up, (viii) checking libraries on the Bioanalyser for expected insert size and concentration (QuantiFluor® dsDNA System kit, Promega), (ix) dilution and mixing, and (x) sequencing, performed by Novogene on a NovaSeq PE250 for individual libraries obtained for 54 diploid green toads (ca. 100 Mio bp raw data per library).

We included Palearctic green toads by the following numbers, taxa and sexes: 8 *B. siculus* (4 ♂♂, 3 ♀♀), 9 *B. viridis* (5 ♂♂, 4 ♀♀), 8 *B. balearicus* (5 ♂♂, 3 ♀♀), 9 *B. shaartusiensis* (4 ♂♂, 5 ♀♀), 10 *B. turanensis* (4 ♂♂, 5 ♀♀). Species' nomenclature follows[36].

The sequence reads from target enrichment were mapped against the reference genome using MINIMAP2[74] (-I 100 G -a -x sr -t 4). SAMTOOLS[83] was used to add read group and sample names, to fix mates and sort the bam file output. All BAM files were merged and variant calling for the target regions was performed by SAMTOOLS and BCFTOOLS[87] with parameters to cope for extreme sequencing depth (samtools view -bu input.bam scf1:324181812-324192250 scf1:566783149-566866298 | bcftools mpileup --annotate DP4 --max-depth 1000000000 -f bufVir1d.fa - | bcftools call -mv -Ob -o calls.bcf). The output was converted to vcf format and summarized by an awk script that filtered SNP position and calculated allele frequency from DP4 annotation for each sample. The resulting table was further inspected in Excel and SNPs that were heterozygous in males while homozygous in females were filtered. To fine map the region of the sex marker male specific SNPs of the different species were intersected (resulting in male specific SNPs present in 2, 3, 4, 5 species) and plotted along the target regions in our genome browser.

### Reporting summary
Further information on research design is available in the Nature Portfolio Reporting Summary linked to this article.

## Data availability
All data generated in this study have been deposited in publicly accessible databases. Genome assembly, whole genome sequencing (WGS) data and RNA-seq reads were submitted to GenBank (Bioproject: PRJNA292872). Alternatively, the IGB-Green Toad genome browser http://genomes.igb-berlin.de/Greentoad/ provides easy access to the assembly and its annotation. GenBank accession numbers used for genome annotation are listed in the Supplementary Text. Source data is provided as a source data file. Source data are provided with this paper.

## Code availability
Scripts and parameters are deposited at https://github.com/HMPNK/BUFVIR.

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

## Acknowledgements

Funding has been in part provided by the German Research Foundation (DFG) STO493/8 to MStöck and HK, and the Leibniz-Institute of Freshwater Ecology and Inland Fisheries (IGB). WKloas and MStöck also received funding from the European Union's Horizon 2020 research and innovation program, grant agreement No. 825753 (ERGO). WHT and CW received funding from the Faculty of Science, National University of Singapore (RSB, A-0008515-00-00). FISH-TSA experiments were partly covered by the PJAC-project CZ.02.01.01/00/22_010/0002902, MSCA Fellowships CZ–UK (MK). We thank Petros Lymberakis, Natural History Museum of Crete, University of Crete, for help with the sampling and export permits of green toads by the Ministry of the Environment of Greece (115790/229) 2014–2017. The tissue and DNA collection of MStöck profited from donations of multiple international colleagues over 25 years; he thanks everyone. We are grateful to Sandra Bittmann (IGB) for HMW-DNA extraction, Eva Kreuz (IGB) for initial efforts to clone the SD-region, Julia Bindl (IGB) for help with the DNA and RNA work, Hughes Parrinello (Plateforme MGX) for preparing the HiC-libraries and sequencing, Susan Mbedi and the BeGenDiv (Berlin, Germany) for providing lab space and Illumina-sequencing support for the ddRAD. We thank Sebastian Petri (Biozyme) for discussions on high-resolution agarose gel-electrophoresis.

## Author contributions

H.K.: contributed to study design, sequenced long reads, assembled and annotated the genome, set up the genome browser, mapped and analyzed pool-seq sequences and SNPs of the AmpliSeq custom DNA panel, performed phylogenetic analyses, contributed to 1st manuscript draft. T.W.H.: made histological sections and performed all RNA in-situ hybridizations and H3K4me immunostainings, contributed to 1st manuscript draft. C.K.: analyzed all transcriptomes, and gene expression data, analyzed splice sites of *bod1l*. WKleiner: Performed RNA extraction, bioanalyzer quantification and evaluation, did most marker testing by PCRs, co-worked in the wet lab for the AmpliSeq custom DNA panel. B.K.: performed molecular work to characterize the ncRNA of multiple green toad taxa, prepared RNA for transcriptomes and cDNA-work at Gosner stage 46. M.C.: contributed to ddRAD wet lab. R.F.: analyzed the ddRADseq data. M.K.: performed FISH-TSA. W.Kloas: contributed to study design, discussed developmental histology and data. M.Schartl: advised in many respects, initiated some collaborations, discussed data, study progress and results. C.W.: contributed to study design, supervised and analyzed RNA in-situ hybridization and H3K4me immunostainings, and contributed to writing. M.Stöck: conceived the study, did all field work/sampling, raised tadpoles, prepared and fixed tissues for chromosome and RNA in situ work, collected buccal swabs and tissues over 25 years, sexed toads, extracted all DNAs, performed ddRAD wet lab, developed all primers, did some marker testing by PCRs, co-worked in the wet lab for the AmpliSeq custom DNA panel, wrote the 1st manuscript draft. All authors contributed text to the manuscript, and read and improved the final version.

## Funding

## Competing interests

The authors declare no competing interests.

## Additional information

[1]Leibniz-Institute of Freshwater Ecology and Inland Fisheries, IGB, Müggelseedamm 301 & 310, 12587 Berlin, Germany. [2]Department of Biological Sciences and Centre for Bioimaging Sciences, National University of Singapore, 14 Science Drive 4, Block S1A, Level 6, Singapore 117543, Singapore. [3]SIGENAE, Plateforme Bio-informatique Genotoul, Mathématiques et Informatique Appliquées de Toulouse, INRAe, 31326 Castanet-Tolosan, France. [4]Department of Molecular Biology and Genetics, Genetics, Faculty of Science, Bilkent University, SB Building, Ankara 06800, Turkey. [5]Danube Delta National Institute for Research and Development, Tulcea 820112, Romania. [6]Advanced Research and Development Center for Experimental Medicine—CEMEX, "Grigore T. Popa", University

of Medicine and Pharmacy, Mihail Kogălniceanu Street 9-13, Iasi 700259, Romania. [7]Department of Ecology and Evolution, University of Lausanne, Lausanne, Switzerland. [8]Swiss Institute of Bioinformatics, Lausanne, Switzerland. [9]Department of Cell Biology, Faculty of Science, Charles University, Viničná 7, Prague 12843, Czech Republic. [10]Department of Biology, McMaster University, 1280 Main Street West, Hamilton, L8S 4K1 Ontario, ON, Canada. [11]Developmental Biochemistry, Biocenter, University of Wuerzburg, Am Hubland, 97074 Wuerzburg, Germany. [12]The Xiphophorus Genetic Stock Center, Department of Chemistry and Biochemistry, Texas State University, San Marcos, TX 78666, USA. [13]These authors contributed equally: Heiner Kuhl, Wen Hui Tan.
✉e-mail: cwinkler@nus.edu.sg; matthias.stoeck@igb-berlin.de

