## [Peer Review File · Nature Communications]

A candidate sex determination locus in amphibians which evolved by structural variation between X- and Y-chromosomesREVIEWER COMMENTS

Reviewer #1 (Remarks to the Author):

The manuscript of Kuhl et al. entitled by "A novel amphibian sex determination candidate gene, evolved by structural variation in the regulatory region between X and Y" describes identification of Y-specific region and the candidate gene of male determination in the European green toad, *Bufo(tes) viridis*, using ddRAD- and whole genome sequencing. The candidate gene is *Bod1L*, of which the coding sequence is not different from each other between homologues of X and Y chromosomes. On the other hand, the non-coding RNA region located and inserted at the upstream region of *Bod1L* is Y specific and has been highly conserved in males only (on Y) through multiple taxa of *Bufo(tes)*. It is noteworthy for publishing this manuscript in this journal, because the Y-specific region identified using whole genome analysis in this work is absolutely evident, which is highly conserved in males only and Y chromosome specific through multiple taxa, and this is the first finding in frogs except *Xenopus laevis*. On the other hand, the expression difference of *Bod1L* between sexes and the relationship with H3K4me seem confusing, and I have some arguments.

1. Y-specificity of the ncRNA on chromosome 1 is evident, but *Bod1L* is unclear in candidacy of sex determining gene. *Bod1L* is expressed in germ cells of both sexes, and the sexual difference of expression in somatic cells at sex-determining stage and thereafter is not clear from fig.2, fig.4 and supplementary figure 10. Even at six weeks after metamorphosis, the expression difference is not significant (fig.2). Could this be a candidate of male determining gene?
2. Likewise, I could not find any positive expression of *Bod1L* in somatic cells of testis in Fig.4d (green arrows).
3. On the other hand, Y-specific ncRNA is rising in expression at stage 5a before metamorphosis while is not expressed in female tissues including gonads. Is this rather the candidate? I mean that the ncRNA could be apart from the enhancer role in *Bod1L* expression, but rather a single male determiner.
4. H3K4 me3 is weaker in somatic cells of males than those of females of this toad (Fig. 4). This is opposite to the case in mammals, where increasing of the H3K4 methylation allows SRY expression activation (male determination). How do the authors discuss this disagreement?
5. This work does not include functional analysis of the candidate gene. Therefore, the title of manuscript referring "candidate" is valid. It is not easy actually to complete functional analysis such as genome editing in frogs other than model amphibians.
Then, I ask you, what about expression of the ncRNA and *Bod1L* in bidder's organ? This organ is a proto-ovary of males and differentiate to ovary after castration. Therefore, Y-specific ncRNA and *Bod1L* in XY male, if they are candidates of male determination, should not be expressed there and even after castration.
6. When does sexually dimorphic expression of *cyp19* and *cyp7* initiate? These two genes are good markers to know the timing of gonadal sexual differentiation in frogs and to estimate the timing of sex-determination. The detection of the two genes expressions is described in the text, but initiation timing of the sexually dimorphic expression is not.

Reviewer #2 (Remarks to the Author):

Kuhl et al.'s study aims to reveal a strong candidate sex-determining gene in a European green toad. There are several major points I wish the authors could consider addressing and provide further clarification.

1. Homomorphy from karyotype alone does not necessarily indicate they do not have genetic differentiation between X and Y, and one cannot assume they might not have any possible gene

gain or loss event unless it has been thoroughly analyzed. Because of the limited detecting power, it is not very convincing the candidate SD would be this one.

a. To do so, it would require performing the coverage ratio (or difference) between females and males along the sex chromosome, to verify whether there is gene loss or Y-specific region evolved.

b. Using heterozygous in males and homozygous in females to detect the potential sex-determining region would miss other possibilities on candidate SD genes, such as Y-limited SNPs or X-limited SNPs. It should include other scenarios to detect these SNPs or associated regions.

c. Because the reference genome is XX females, so if there is Y-specific genes/regions lacking in females, it will be missed. It would require detecting the orphan reads from mapping XY males against the XX genome, to assemble the Y-specific regions.

2. In all studied amphibians that males have telomere-restricted recombination across the genome, so that it is expected to find high density of SNPs in males throughout the majority of sex chromosome. This is consistent with the RAD-seq approach detected sex-linked markers across the entire sex chromosome (Chr01), but somehow this is contradictory as the whole-genome pool-seq data results (Fig.1d). One would expect to detect elevated F_{st} between sexes along the sex chromosome given the extreme heterochiasmy recombination pattern. How are the two approaches in the same study not consistent with each other? How do the authors interpret the discrepancy of the two results?

3. Given the sex-linked markers were also detected (by both RADseq and pool-seq) on Chr07, and Chr07 is proportionally much smaller so that less sex-linked SNPs are expected. It would be convincing to provide the genome wide F_{st} scan, homozygosity/heterozygosity etc. figure. One might think it is a possibility two chromosomes co-segregate with sex in some populations perhaps.

4. It is also shown that (homomorphic) sex chromosome polymorphism within one species or even within one population is not uncommon in Ranidea true frogs, which makes it difficult to detect sex chromosomes if sample size from one population is not big or too much variation is induced by sampling from massive number of populations. What do we know about European green toad? For this pool-seq or pool-sex approach here in this study, almost each of the >20 females and >20 males are from different population across the species distribution range. This would introduce huge genetic differentiation across the samples even within each sex. I wonder how this affect the detected results.

5. The results on 5 stages of developmental gene expression are not convincing to suggest the pattern for candidate SD gene.

a. Firstly, it is possible that the authors might completely miss the true SD period, given the sampling stages. It is not right to state that "...differentiation is histologically detectable only at or after metamorphosis completion (>Gosner 46)", both citations for this one is not relevant for this statement. Various histology studies in frogs show early detection of sexual differentiation of gonad via histology is between Gosner 25 to 28 depending on the species (Ogielska et al. J Morphol. 2004; Haczkiwicz et al. Zool Sci. 2013; among others). The gonad differentiation can be detected under standard dissection microscope at or after metamorphosis.

b. Secondly, since the frog's development speed can be affected on many factors, so please indicate the rearing condition including temperature, light:dark cycle, humidity, density per clutch/per group, feeding scheme etc. Thus 10 days after fertilization is unclear that the developmental stage is. Which Gosner stage is this earliest stage? And other few stages only indicating by the days after fertilization needs to specify the Gosner developmental stage.

c. Thirdly, the conclusion based on the limited and unclear developmental stages samples were taken is not very strong, "...we consider these slight quantitative differences between males and females as sufficient to trigger green toad sex determination". It is very possible that the candidate SD has a narrow high sex-biased or sex-specific gene expression window (i.e. a couple of hours or even 1-2 days) which the study missed the sampling period. This conclusion does not support this gene is a candidate SD.

6. It would be good to Blast the entire genome to check the possible (partial) duplication of the detected candidate SD gene and the promotor/enhancer region, given the ≥ 3 bands were detected in males across various $2n$ Bufo species (Fig.1e). The author did the FISH to verify duplication, there might be possibility of missing partial duplication or due to technical limit.

7. Figure 1e and 1h, data is unclear and color code is misleading.

a. In 1e: for species ii, the size of the single band in female is not matching any of the two bands detected in males, what happened to this species?

b. In 1e: for species iii, iv, v, and vi, there are often $2n$ but three or more various-size bands

detected in males. What are these? Can there be duplications in other regions across the genome?
c. The color code of 1ei, ii, iii, iv, v, vi does not match the color code in 1h. 1h the color code in ploidy level and species level are confusing. Please check and clarify.
8. The code/scripts employed for the analysis of diverse results and the generation of figures should be deposited in a public repository, thereby enhancing the reproducibility of results.

Minor comments below:

1. Various points suggest the authors should do a more careful check to avoid draft version related format issue.
 - a. line 66, '(incl. refs)' should be deleted from the draft version.
 - b. Line 92, '(refs. therein)' should be deleted.
2. Title: 'sex determination candidate gene' sounds weird, change to 'candidate sex-determining gene' or 'candidate sex determination gene'.
3. Line 38: master gene, better change to 'master sex-determining gene'.
4. Line 59, the statement only focus on vertebrates is a bias, please add invertebrate lineages, plants etc. to better reflect the true nature of homomorphic sex chromosome distribution across the tree of life.
5. Line 69, the statement on 95% of amphibians possess undifferentiated sex chromosomes is outdated, given the reference was back to Schmid 2001 cytogenetic studies. Later on in particular after 2010, Schmid published various cytogenetic books to include many more heteromorphic sex chromosomes in various frog lineages. Please keep this updated.
6. Line 117, 'N50-11.6 kbp' must be a mistake, 11.6 kbp is even not the read length for Nanopore.
7. Line 125, please elaborate a bit more on the analysis to detect 68.5% of genome size is TEs, just 'De novo analysis' is unclear.
8. Line 141, given 26 detected sex-linked markers were not aligned uniquely or with low quality, this should be already filtered out as not good markers for downstream analysis. Please clearly indicate the SNP/marker filtering strategy.
9. Line 147, please be consistent with the term 'pool-seq' and 'pool-sex'. If using pool-sex, please clarify this term the first use of it.
10. Line 152, given this is the most important result to detect the candidate SD region, the statement here needs to be more elaborated. How mapping pool-sex short reads can reveal a single peak in a region? The heterozygous SNPs (N=105) is clearly a window size, because there were in total 22 males. Although the M&M lists a bit more technical details, it is good to describe the biological reasons and summarize the main criteria used to detect the candidate SD region.
11. Figure 1a, b, c should not be overlap with each other, also figure 1a do not show the X and Y axis, which one is the reference genome which one is the focus one? Figure 1d is missing X and Y axis. Figure 1h the legend is too small and resolution is not good to read clearly, also the color scheme between ploidy level and species should be clarified.
12. Reference 1-54 and the rest of reference were cut into two parts, in between inserted Online content such as Methods etc. Please make sure the format is clear and put all reference together.
13. Line 787, explain MS222 and which percentage was used for adults and different developmental stage.
14. Line 814: what is 'spreadsheet software'? Also please check the Methods section to put all related citations on various software used for analysis, several of them were missing.

RESPONSE TO REVIEWERS' COMMENTS

Reviewer #1 (Remarks to the Author):

The manuscript of Kuhl et al. entitled by “A novel amphibian sex determination candidate gene, evolved by structural variation in the regulatory region between X and Y” describes identification of Y-specific region and the candidate gene of male determination in the European green toad, *Bufo(tes) viridis*, using ddRAD- and whole genome sequencing. The candidate gene is *Bod1L*, of which the coding sequence is not different from each other between homologues of X and Y chromosomes. On the other hand, the non-coding RNA region located and inserted at the upstream region of *Bod1L* is Y specific and has been highly conserved in males only (on Y) through multiple taxa of *Bufo(tes)*. It is noteworthy for publishing this manuscript in this journal, because the Y-specific region identified using whole genome analysis in this work is absolutely evident, which is highly conserved in males only and Y chromosome specific through multiple taxa, and this is the first finding in frogs except *Xenopus laevis*. On the other hand, the expression difference of *Bod1L* between sexes and the relationship with H3K4me seem confusing, and I have some arguments.

Response: We thank the reviewer for this generally positive perception of our manuscript.

1. Y-specificity of the ncRNA on chromosome 1 is evident, but *Bod1L* is unclear in candidacy of sex determining gene. *Bod1L* is expressed in germ cells of both sexes, and the sexual difference of expression in somatic cells at sex-determining stage and thereafter is not clear from fig.2, fig.4 and supplementary figure 10. Even at six weeks after metamorphosis, the expression difference is not significant (fig.2). Could this be a candidate of male determining gene?

Response: We thank the reviewer for this comment and agree that in the previously used Fig. 4, *bod1l*-expression in somatic cells of male gonads was not entirely clear in the images provided. To address this point, we have now re-arranged Fig. 4 and provide higher magnification zoom-in views that better show expression of *bod1l* in the somatic cells (green arrows in Figs. 4b and c) that surround the primordial germ cells (labelled by magenta asterisks) of male gonads at Gosner stage 36-37. We did not observe such expression in somatic cells of female gonads, nor in somatic cells of male gonads at a later stage (Gosner 43-44). For better clarity, images showing *vasa*-expression have been moved to the revised Supplementary Fig. 10. It shows overview images of the analyzed samples and provides additional examples for *bod1l*-expression in male somatic cells at Gosner stage 36-37. Based on these findings, we propose that the first difference observed is that *bod1l* is expressed in somatic cells exclusively of the male but not the female gonad at a critical stage of sex determination (Gosner stage 36-37).

We hypothesize *bod1l* acts on methylation, and even small expression changes could have huge effects and change regulation of SD genes. According to our RNA *in-situ* analysis, the changes take place in the pre-Sertoli cells only. This makes it extremely hard to detect this signal using RNAseq of the whole gonad.

Subsequently, the second differences observed by RNAseq from trunk sections are detected at Gosner stage 43-44), when the ncRNA-Y increases its expression, and the Y-specific *bod1l*-allele is higher expressed than the X-specific one. At the same stage, expression of the ncRNA-Y becomes elevated, suggesting transcriptional changes in the whole region. Importantly, at the same time (in parallel), the additive *bod1l*-expression is from this stage onwards stronger in males than in females.

2. Likewise, I could not find any positive expression of *Bod1L* in somatic cells of testis in Fig.4d (green arrows).

Response: As mentioned above, Fig. 4 has been changed to more clearly show expression of *bod1l* in somatic cells of the male gonad at Gosner stage 36-37.

3. On the other hand, Y-specific ncRNA is rising in expression at stage 5a before metamorphosis while is not expressed in female tissues including gonads. Is this rather the candidate? I mean that the ncRNA could be apart from the enhancer role in *Bod1L* expression, but rather a single male determiner.

Response: We agree with the reviewer and have now considered this option. Indeed, the ncRNA-Y exhibits a level of expression which is an order of magnitude lower than that of *bod1l* (Fig. 2). The ncRNA-Y sits in the pre-genic region of *bod1l* and is co-expressed with it.

To address this comment, we call our finding a “candidate sex determination locus” throughout the manuscript. Since the reviewer has proposed an interpretation of the ncRNA as the potential master gene, we have now also integrated this idea into the Discussion.

4. H3K4 me3 is weaker in somatic cells of males than those of females of this toad (Fig. 4). This is opposite to the case in mammals, where increasing of the H3K4 methylation allows SRY expression activation (male determination). How do the authors discuss this disagreement?

Response: The reviewer is right. However, we actually do not claim that the situation in green toads is identical to that in mammals and therefore called it “reminiscent”. As a major difference, the bipotential primordial gonad in bufonid toads includes the “ovary-like” Bidder’s organ which, however, has an unknown evolutionary origin and function in males (see also response to point 5). To address this point of the referee, we have now inserted two sentences discussing this point:

“However, opposite to the situation in mammals, H3K4me3 is weaker in somatic cells of green toad males than those of females (Fig. 4), while an increase in H3K4-methylation allows Sry-expression activation (male determination). Another major difference is that the primordial gonad of bufonid toads, specifically in green toad males, includes the “ovary-like” Bidder’s organ.”

We also assume that the differential methylation has an effect possibly regardless of its direction.

5. This work does not include functional analysis of the candidate gene. Therefore, the title of manuscript referring “candidate” is valid. It is not easy actually to complete functional analysis such as genome editing in frogs other than model amphibians.

Response: We agree that a functional analysis would go beyond the scope of the present study.

Then, I ask you, what about expression of the ncRNA and *Bod1L* in bidder’s organ? This organ is a proto-ovary of males and differentiate to ovary after castration. Therefore, Y-specific ncRNA and *Bod1L* in XY male, if they are candidates of male determination, should not be expressed there and even after castration.

Response: This is a challenging question since the evolutionary nature of Bidder’s organ is still under debate (e.g. Piprek *et al.* 2014).

Two important differences between common toads (*Bufo bufo*), in which Harms (1923, now cited) and Ponse (1941) made their “classical” sex reversal experiments, and green toads (*B. viridis*) should be taken into account: (i) the *B. bufo* of Ponse was concluded to be female heterogametic (ZZ/ZW), whereas *B. viridis* exhibits a male heterogametic system (XX/XY). (ii) Both Harms (1923) and Ponse (1941) castrated adult male *B. bufo* for sex reversal, while we analyzed gene expression in developing juvenile organs. Several authors consider inhibition of Bidder’s organ’s development by the development of mature/-ing testes (discussed by Piprek 2014).

Importantly, we observed anatomical differentiation into Bidder’s organ and testis in males, and ovaries in females, in our Gosner stage 46 (some toads 6 weeks after metamorphosis) in juvenile green toads. At this stage, the Y-specific ncRNA and *bod1l* in XY-males are expressed in both organs (and we did not perform castration experiments either in juvenile or adult toads). To address this point, we added information on anatomical differentiation at Gosner stage 46 to the legend of Fig. 2 (“Gosner stages 45-46, up to: six weeks after metamorphosis, the latter

with anatomically visible differentiation of testes and Bidder's organs in males, and ovaries in females,...") and to the Discussion:

"The expression of the ncRNA-Y and *bod11* in Bidder's organ of the differentiating male gonad (Gosner stage 46) may contradict the view that Bidder's organ in this XX/XY system can be simply interpreted as a 'proto-ovary'."

6. When does sexually dimorphic expression of *cyp19* and *cyp7* initiate? These two genes are good markers to know the timing of gonadal sexual differentiation in frogs and to estimate the timing of sex-determination. The detection of the two genes expressions is described in the text, but initiation timing of the sexually dimorphic expression is not.

Response: We assume that *cyp17* (?) and *cyp19* are referred to here, which are both expressed at Gosner stage 46 and later on. This is shown in Suppl. 5 and described in the main text: "Expression of *cyp19a1a*, *cyp17a1* and *gdf9* was detectable only in Gosner 46 and adult ovaries."

To further address this, we added to the relevant sentence "with *cyp19a1a* and *cyp17a1* presenting good markers for the timing of gonadal sexual differentiation in anurans".

Reviewer #2 (Remarks to the Author):

Kuhl et al.'s study aims to reveal a strong candidate sex-determining gene in a European green toad. There are several major points I wish the authors could consider addressing and provide further clarification.

1. Homomorphy from karyotype alone does not necessarily indicate they do not have genetic differentiation between X and Y, and one cannot assume they might not have any possible gene gain or loss event unless it has been thoroughly analyzed. Because of the limited detecting power, it is not very convincing the candidate SD would be this one.

Response: In response, we have now assembled, in addition to the female assembly, a whole genome assembly of a male (submitted to GenBank at the same bioproject) and analysed the pool-seq coverage as proposed. Also, we have compared both assemblies to (re-)establish the quantity of heterozygous SNPs. All three approaches referred to the same locus in the chromosomal-level assemblies (Fig. 1c), each yielding the highest peak (see also below).

a. To do so, it would require performing the coverage ratio (or difference) between females and males along the sex chromosome, to verify whether there is gene loss or Y-specific region evolved.

Response: We did poolseq coverage analysis on the female genome in the beginning, but this did not give clear results. Generally, 'noisy' poolseq coverage and genomes with high amounts of repetitive elements are problematic. In such a case, small regions with haploid coverage differences are barely or not clearly detectable, while megabase-sized regions would be detectable (then 'noise' can be reduced by using average values across large genome windows for analysis). We did not find such larger regions with haploid coverage difference between male and female poolseq data using our female *B. viridis* reference genome.

To address this comment, we have therefore *de novo*-assembled a male genome, and have repeated these analyses. Likewise, we did not find any larger (megabase-sized) regions with haploid coverage difference in the male genome assembly. Nevertheless, we were able to detect smaller regions with male-specific poolseq-coverage, again, with the strongest signal on the assembled chromosomes in the ncRNA-Y/*bod11* region (now shown for scaffold 1 in Figure 1c). Male long-read data of a single individual supported this haploid Y-specific region (this information is now included in the manuscript). None of the other top 200 regions with male-specific poolseq coverage had any overlap with 'usual suspect' SD-genes, but only the candidate locus *bod11*//ncRNA-Y turned out to be sex-daignostic, both intra-specifically as well as inter-specifically. Overall, the detection power (strongest signal vs. second best signal) was still better with the male-specific heterozygous SNP approaches, which are now available for both male and female assemblies (Fig. 1c). Thus, we now provide three different analyses

that highlight the same locus on scf1/chr1 as different between X and Y chromosomes. These three independent analyses now described in the revised manuscript disclosed a marker that can even discriminate between sexes of different *Bufo* taxa, which strongly supports our datamining results.

b. Using heterozygous in males and homozygous in females to detect the potential sex-determining region would miss other possibilities on candidate SD genes, such as Y-limited SNPs or X-limited SNPs. It should include other scenarios to detect these SNPs or associated regions.

Response: As described above, we have now made major efforts and present an additional full *de novo* male genome assembly. We also modified Figure 1 (now 1c) and added new analyses of the male genome assembly. We show the signals for male-specific pool-seq coverage analysis on the male genome, as well as analyses for male heterozygous SNPs on scf1/chr1 next to the previous female reference genome-based study to underline their strong overlap. The comprehensive analyses for the entire female and male genomes are now added as Supplementary File 2.

c. Because the reference genome is XX females, so if there is Y-specific genes/regions lacking in females, it will be missed. It would require detecting the orphan reads from mapping XY males against the XX genome, to assemble the Y-specific regions.

Response: Detecting orphan reads from mapping XY males against the XX genome was one of our initial approaches, but did not allow us to assemble sequences of sufficient continuity. As typical of repeat-rich large amphibian genomes, such analyses can become extremely difficult, as potentially Y-specific reads might still map to similar regions in the female genome and are lost. This is why we did prefer a *de novo* assembly approach of the male genome to address this reviewer's comment. As stated above, we added the male assembly and re-analyzed the data, confirming the previous results.

2. In all studied amphibians, that males have telomere-restricted recombination across the genome, so that it is expected to find high density of SNPs in males throughout the majority of sex chromosome. This is consistent with the RAD-seq approach detected sex-linked markers across the entire sex chromosome (Chr01), but somehow this is contradictory as the whole-genome pool-seq data results (Fig.1d). One would expect to detect elevated Fst between sexes along the sex chromosome given the extreme heterochiasmy recombination pattern. How are the two approaches in the same study not consistent with each other? How do the authors interpret the discrepancy of the two results?

Response: Recombination between putative XY sex chromosomes has been well-studied in Palearctic green toads using microsatellites and sequence markers. These previous data showed reduced X-Y recombination in males, while phylogenetic analyses of sex-linked sequences show that X and Y alleles clustered by species, not by gametolog. This led to the conclusion that X-Y homomorphy and fine-scale sequence similarity in these species do not stem from recent sex-chromosome turnovers, but from occasional X-Y recombination (Stöck et al. 2013, doi: 10.1111/jeb.12086; Tamschick et al. 2015, doi:10.1159/000380841).

We had referred to this situation in the Introduction: "All feature strongly-reduced X/Y-recombination but occasional X/Y-exchange is suggested at evolutionary time scales²⁹."

This seems fully consistent with and reconciling the ddRAD data stemming from three genetic families and the poolsex data originating from different range parts of *B. viridis*, probably governed by occasional X-Y recombination during evolutionary time scales (see also our response to point 4, below).

Recombination of scf1 is still happening (we can observe its signatures at the borders of the Y-specific haplotype between species, see Suppl. Figure 4a). Sex-linked SNPs over the whole chromosome can be found in genetic families (our RADseq data). As soon as different populations are considered (our poolseq data), the sex-specific signal narrows down to the diagnostic SD-region. Therefore, we consider data from only a few genetic families to perform insufficiently to reveal a tiny SD- locus. We have also added to the manuscript: "and sex-

linkage of most ddRAD markers is not evolutionarily conserved due to X-Y recombination, in the huge pseudo-autosomal region (PAR) of this chromosome 1”.

3. Given the sex-linked markers were also detected (by both RADseq and pool-seq) on Chr07, and Chr07 is proportionally much smaller so that less sex-linked SNPs are expected. It would be convincing to provide the genome wide Fst scan, homozygosity/heterozygosity etc. figure. One might think it is a possibility two chromosomes co-segregate with sex in some populations perhaps.

Response: On scf7, a single marker was detected exclusively in RADseq. This scaffold is about 25% the size of scf1. On scf1, the linkage group identified earlier as the sex chromosome in green toads (e.g. Brelford et al. doi:10.1111/evo.12151), we detected 64 RADsex markers. This means about 16-fold higher density of RADseq markers on scf1 compared to scf7/chr7. We consider this single marker on chr7 as a false positive, because it had a mapping quality (MQ) only slightly above our cut-off (cut-off MQ = 20; scf7 marker MQ = 22). If we use the highest possible / most stringent cut-off MQ = 60, those RADsex markers only map to scf1 / chr1. Several independent marker types in several previous papers have provided evidence for scf1 to be the sex chromosome. Regarding the proposal, we do now provide genome wide plots of male poolsex specific heterozygous SNPs in Supplementary File 2 for both genome assemblies.

4. It is also shown that (homomorphic) sex chromosome polymorphism within one species or even within one population is not uncommon in Ranidea true frogs, which makes it difficult to detect sex chromosomes if sample size from one population is not big or too much variation is induced by sampling from massive number of populations. What do we know about European green toad? For this pool-seq or pool-sex approach here in this study, almost each of the >20 females and >20 males are from different population across the species distribution range. This would introduce huge genetic differentiation across the samples even within each sex. I wonder how this affect the detected results.

Response: We respectfully disagree. We have used prior phylogeographic knowledge on green toads to ensure that only the same taxa (species or subspecies) that also share scaffold 1 as the sex chromosome were included into the pool-sex approach.

As we did so, it may be counter-intuitive, but inter-population variation actually clearly increases the chance to find a sex-specific locus, as signals of “sex-linked” variants (i.e. potential false positives) are diminished, while truly sex-diagnostic signals are retained. This actually made it possible to find the sex-specific locus and not just “re-find” the sex chromosome. This approach has shown its power in several systems, especially of teleosts. To address this comment, we have added a sentence to explain the informed choice of these individuals: “chosen to be closely related according to phylogeographic information (refs. 40,41))and presumably sharing the same sex chromosome based on microsatellites (refs.33-35)” to the manuscript.

5. The results on 5 stages of developmental gene expression are not convincing to suggest the pattern for candidate SD gene.

a. Firstly, it is possible that the authors might completely miss the true SD period, given the sampling stages. It is not right to state that “...differentiation is histologically detectable only at or after metamorphosis completion (>Gosner 46)”, both citations for this one is not relevant for this statement. Various histology studies in frogs show early detection of sexual differentiation of gonad via histology is between Gosner 25 to 28 depending on the species (Ogielska et al. J Morphol. 2004; Haczkiwicz et al. Zool Sci. 2013; among others). The gonad differentiation can be detected under standard dissection microscope at or after metamorphosis.

Response: We assume this comment might be based on a misunderstanding.

As this is a protected species, for the earlier stages, we have included samples of control groups (untreated tadpoles from a previous) experiment, in which we initially had no Gosner

stages recorded. We have now used frozen tadpoles from this experiment, transferred them to 70% ethanol, determined the Gosner stages *à posteriori*, and adapted the text throughout the manuscript.

We have sampled the larvae from early after the formation of the operculum (ca. 10 days after fertilization), i.e. consecutively from ca. Gosner stage 23 to 46 (plus metamorphosed Gosner 46-toads at 6 weeks after metamorphoses).

In any case, as shown in the literature and by some of us (e.g. Tamschick et al. 2014) and other authors (e.g. Piprek et al. 2014), there is no doubt that histologically detectable sexual differentiation occurs in *B. viridis* different from the studies referred to by this reviewer, from Gosner stage 46 and thereafter. Therefore, we can say with confidence that the genetic sex determination period lies before this stage and is fully covered by our developmental time series.

b. Secondly, since the frog's development speed can be affected on many factors, so please indicate the rearing condition including temperature, light:dark cycle, humidity, density per clutch/per group, feeding scheme etc.

Response: We agree that this is important and have inserted detailed information in the M&M part.

Thus 10 days after fertilization is unclear that the developmental stage is. Which Gosner stage is this earliest stage? And other few stages only indicating by the days after fertilization needs to specify the Gosner developmental stage.

Response: Gosner stages from 23 to 46 have now been added and the text adapted throughout.

c. Thirdly, the conclusion based on the limited and unclear developmental stages samples were taken is not very strong, "...we consider these slight quantitative differences between males and females as sufficient to trigger green toad sex determination". It is very possible that the candidate SD has a narrow high sex-biased or sex-specific gene expression window (i.e. a couple of hours or even 1-2 days) which the study missed the sampling period. This conclusion does not support this gene is a candidate SD.

Response: (i) We present a genome-wide marker that allows diagnostic PCR-based sexing of *B. viridis* and several related taxa, (ii) as expected, its locus phylogenetically clusters by gametologue and not by species; (iii) we show sex-specific (exclusively male) expression of the ncRNA-Y, and (iv) sex-biased expression of *bod1l* in primordial gonads, prior and coinciding with histologically visible differentiation (Gosner stages 27-46, plus 6 weeks after metamorphosis), (v) a stronger male additive *bod1l*-expression from Gosner stage 43-44. All of this cumulative evidence (i-v) should be evaluated together. In conclusion, we are sure that we did not miss the relevant gene expression in >50 transcriptomes.

Beyond this cumulative evidence, if we follow the reviewer's scenario, assuming there may even be a higher developmental peak of sex-specific expression of the candidate gene/s, which unfortunately has not been caught because of potentially missing stages, we think that the other above evidence still points to the right genes.

6. It would be good to Blast the entire genome to check the possible (partial) duplication of the detected candidate SD gene and the promotor/enhancer region, given the ≥ 3 bands were detected in males across various 2n Bufo species (Fig.1e). The author did the FISH to verify duplication, there might be possibility of missing partial duplication or due to technical limit.

Response: We have shown the duplication in the haplotype X versus haplotype Y dot plot figures (Supplementary Figure 7) and now by addition of the male WGS (Supplementary File 2), which explains that they stem from the same genomic region and not from elsewhere in the genome.

Additional tandem duplications of the marker in the other species, sharing the marker, are possible (likely), but beyond the framework of this paper, since more long-read WGS would be needed for these species.

7. Figure 1e and 1h, data is unclear and color code is misleading.

a. In 1e: for species ii, the size of the single band in female is not matching any of the two bands detected in males, what happened to this species?

Response: We agree that the highest band of the male in this species is not easy to see. We have provided this gel with the same PCR conditions for all taxa involved. We added a note to the figure legend. We also successfully amplified the genomic sequence of the ncRNA-Y only from males of this taxon (as used for the tree in Figure 3).

b. In 1e: for species iii, iv, v, and vi, there are often 2n but three or more various-size bands detected in males. What are these? Can there be duplications in other regions across the genome?

Response: We agree this situation should be better discussed. We have now added: “two to three bands in males, widely consistent with an XX♀/XY♂-system, but suggesting multiple Y-specific signals in males, which might be sex-chromosomal duplications, nonspecific PCR-amplicons or even PCR-recombinants.”

c. The color code of 1ei, ii, iii, iv, v, vi does not match the color code in 1h. 1h the color code in ploidy level and species level are confusing. Please check and clarify.

Response: We apologize for this error; this has been now corrected.

8. The code/scripts employed for the analysis of diverse results and the generation of figures should be deposited in a public repository, thereby enhancing the reproducibility of results.

Response: For this, we have set up a github repository which will be made available upon acceptance of the manuscript (<https://github.com/HMPNK/BUFVIR>).

Minor comments below:

1. Various points suggest the authors should do a more careful check to avoid draft version related format issue.

a. line 66, '(incl. refs)' should be deleted from the draft version.

b. Line 92, '(refs. therein)' should be deleted.

Response to a. and b.: Both deleted. However, we had actually tried to show that we are fully aware of presenting only a minimum number of references due to space limitations and to show we are aware of the complex evidence at these two very occasions.

2. Title: 'sex determination candidate gene' sounds weird, change to 'candidate sex-determining gene' or 'candidate sex determination gene'.

Response: Changed acc. to 1st suggestion and adapted.

3. Line 38: master gene, better change to 'master sex-determining gene'.

Response: Changed; we now call this a 'candidate sex-determining locus'.

4. Line 59, the statement only focus on vertebrates is a bias, please add invertebrate lineages, plants etc. to better reflect the true nature of homomorphic sex chromosome distribution across the tree of life.

Response: We think this is truly beyond the scope of our paper. Nevertheless, we now refer to their prevalence in the tree of life and cite a textbook.

5. Line 69, the statement on 95% of amphibians possess undifferentiated sex chromosomes is outdated, given the reference was back to Schmid 2001 cytogenetic studies. Later on in particular after 2010, Schmid published various cytogenetic books to include many more heteromorphic sex chromosomes in various frog lineages. Please keep this updated.

Response: We are fully aware of these papers (e.g. Stöck et al. 2021: <https://doi.org/10.1098/rstb.2020.0426>) but were limited in the number of references for the 1st journal (ca. 50!). We adapted the relevant sentences and added another reference referring to several examples of heteromorphic sex chromosomes in amphibians, also co-authored by one of us.

6. Line 117, 'N50-11.6 kbp' must be a mistake, 11.6 kbp is even not the read length for Nanopore.

Response: No this is not a mistake. ONT-sequencing typically delivers N50 read length in the range of 10 - 20 kbp for standard ligation sequencing (meaning 50% of the sequencing coverage is provided by reads longer than this value). Nevertheless, our data contain some fraction of longer reads +20 kbp of course, which are highly beneficial for genome assembly. Read-length N50 larger 50 kbp can be obtained from protocols for ultra-long library sequencing, but highly reduces the total sequencing yield in terms of sequencing coverage.

7. Line 125, please elaborate a bit more on the analysis to detect 68.5% of genome size is TEs, just 'De novo analysis' is unclear.

Response: This value is the total amount of repetitive sequence found in our assembly. It does not just count TEs, but also low complexity sequences, satellite DNA and so on. It was obtained from RepeatModeler/Repeatmasker *de novo* analysis of our female genome assembly. We have added this to the manuscript: "*De novo* repeat analysis of the assembly was performed by the RepeatModeler / RepeatMasker package".

8. Line 141, given 26 detected sex-linked markers were not aligned uniquely or with low quality, this should be already filtered out as not good markers for downstream analysis. Please clearly indicate the SNP/marker filtering strategy.

Response: The RADsex tool, written by our co-author R. Feron does not apply SNP calling, but identifies sex-specific RAD sequences. Parameters (mainly default) used for this tool are in the methods part. The identified sex-specific RAD sequences were mapped against the male and female genomes and here we filtered for mapping quality ≥ 20 . We have added "mapping quality (MQ)" to the text to underline this filtering step, mapping against both genomes revealed nearly identical results.

9. Line 147, please be consistent with the term 'pool-seq' and 'pool-sex'. If using pool-sex, please clarify this term the first use of it.

Response: We actually had explained the term in the previous version in parentheses; to address this, we have now added ("from here: 'pool-sex' ").

10. Line 152, given this is the most important result to detect the candidate SD region, the statement here needs to be more elaborated. How mapping pool-sex short reads can reveal a single peak in a region? The heterozygous SNPs (N=105) is clearly a window size, because there were in total 22 males. Although the M&M lists a bit more technical details, it is good to describe the biological reasons and summarize the main criteria used to detect the candidate SD region.

Response: Yes, our approach is similar to a window approach, but we do not use fixed window sizes, which could be suboptimal if a region of dense SNPs is distributed over two or more windows. Instead, we cluster the heterozygous male-specific variants, if they are close to each other in the genome (i.e. < 9 kbp in our case).

11. Figure 1a, b, c should not be overlap with each other, also figure 1a do not show the X and Y axis, which one is the reference genome which one is the focus one?

Response: "x-axis" for: *B. bufo* and "y-axis" for *B. viridis* has been added to the legend.

Figure 1d is missing X and Y axis.

Response: We have now added this information to the Figure legend.

Figure 1h the legend is too small and resolution is not good to read clearly, also the color scheme between ploidy level and species should be clarified.

Response: Fig. 1 has been newly arranged.

12. Reference 1-54 and the rest of reference were cut into two parts, in between inserted Online content such as Methods etc. Please make sure the format is clear and put all reference together.

Response: This was a transfer manuscript and references were formatted according to the 1st journal. This has been corrected.

13. Line 787, explain MS222 and which percentage was used for adults and different developmental stage.

Response: We added the following description: "Tricaine PHARMAQ 1000 MG/G (MS222)" is a commonly used veterinary drug". To address this, we have added "(and two adult)" and "(concentration: 500 mg/L)" in the M&M part.

14. Line 814: what is 'spreadsheet software'? Also please check the Methods section to put all related citations on various software used for analysis, several of them were missing.

Response: We specified this as "Microsoft Excel".

REVIEWERS' COMMENTS

Reviewer #1 (Remarks to the Author):

Dear Authors,

Even though I am not sure from the experimental results whether the Y-specific ncRNA is the sex determining gene itself or the enhancer of *Bod1l* expression in XY males or their roles in male determination, it is noteworthy that this study definitely identified the male-determining locus in this species and probably is conserved in some of the related species. This work suggests that a sex determining gene in frogs can evolve by genomic arrangements in the upstream region of a sex determining candidate gene, and that's why identification of the sex determining gene has not been easy so far in frogs except *Xenopus laevis*. Therefore, I strongly recommend publication of this manuscript, and hope that the sex determining gene in the green toad will be functionally proved and identified in future by the work of the authors using transgenesis and genome editing. I believe that it can be done even in toads though taking a time.

Reviewer #3 (Remarks to the Author):

I am reading this MS for the first time, but I see that in the previous round of review the other reviewers gave very insightful comments, to which the authors have responded carrying out a great amount of additional work to address the questions raised. The addition of a full genome assembly of a male toad, and also the addition of an extensive transcriptomic analysis of tadpoles spanning most of the larval period of the species seems to have strengthened the support for the initial findings reported. At this stage, I think that the combined evidence derived from the chromosome-level genome assemblies provided, the transcriptomic analyses, the whole-genome pool-seq, ddRAD-seq and targeted sequencing are congruent enough to allow the authors to confidently describe the identified locus as a candidate sex determination one, only pending confirmation through downstream functional analyses. The authors seem to have also extensively revised the text from the previous version and seem to have addressed most of the previous comments. I therefore only have additional minor comments/suggestions to add at this point:

L. 61-62. Please use 'reptile' rather than 'reptilian vertebrate' and do not refer to poikilotherms as cold-blooded, as it is an old concept and rather inaccurate.

L. 97-100. This sentence is a bit long and hard to follow. Would this other structure work better? 'Molecular evidence for male heterogamety resulted from sibship analyses using microsatellites and nuclear sequence markers (refs), identified the largest linkage group (LG1) as sex-linked in several diploid green toad species. This linkage group is homologous to the autosomal LG1 of *X. tropicalis* and harbours the gene *dmrt1*.'

L. 100-102. This sentence is not very clear either. Do you mean that all the green toad species show strongly reduced X/Y recombination but show some X/Y exchange in the pseudo-autosomal region over evolutionary timescales?

L. 115. The additional species included in the study are not revealed until page 24, at the end of the Methods section. I think it should be indicated in the introduction which taxa were studied, and most definitely at least when you report results on them (e.g. targeted enrichment analyses on L. 209).

Also, when referring to the *Bufo* species you should use a recently updated species nomenclature, unless you have specific reasons not to do it. You indicate (L. 800) that you follow Ueda 1990 but perhaps it would be more appropriate to use the recent Dufresnes, C., Mazepa, G., Jablonski, D., Oliveira, R. C., Wenseleers, T., Shabanov, D. A., ... & Litvinchuk, S. (2019). Fifteen shades of green: the evolution of *Bufo* toads revisited. *Molecular Phylogenetics and Evolution*, 141, 106615.

L. 117 X and Y chromosomes?

L. 139. The three distinct genetic families are not clearly described in the methods. You refer to one set of samples consisting of the two parent toads and 40 of their offspring, which I take as one genetic family, and then two groups of 25 or 15 siblings. Are these the two other genetic families? I take then that these two groups of siblings came from two additional pairs of adult toads? Please clarify.

L. 171. State the number and origin (e.g. tissue, larval or adult) of these transcriptomes and perhaps refer to the corresponding Methods section.

L. 533. 'consisting of'. Also change to '40 offspring sexed mostly phenotypically'. How do you 'mostly' sex them phenotypically? Were some ambiguous? Did you use additional methods to sex them, other than phenotypically?

L. 260. Delete extra parenthesis.

L. 362. Our data on *B. viridis*...

L. 497. Can you provide more detail as to the annotation procedure? Which RNA-Seq data were used? What annotation pipeline did you use?

RESPONSE TO REVIEWERS' COMMENTS

Reviewer #1 (Remarks to the Author):

Dear Authors,

Even though I am not sure from the experimental results whether the Y-specific ncRNA is the sex determining gene itself or the enhancer of *Bod1l* expression in XY males or their roles in male determination, it is noteworthy that this study definitely identified the male-determining locus in this species and probably is conserved in some of the related species. This work suggests that a sex determining gene in frogs can evolve by genomic arrangements in the upstream region of a sex determining candidate gene, and that's why identification of the sex determining gene has not been easy so far in frogs except *Xenopus laevis*. Therefore, I strongly recommend publication of this manuscript, and hope that the sex determining gene in the green toad will be functionally proved and identified in future by the work of the authors using transgenesis and genome editing. I believe that it can be done even in toads though taking a time.

Response: Thanks for the very positive perception of our revision.

Reviewer #3 (Remarks to the Author):

I am reading this MS for the first time, but I see that in the previous round of review the other reviewers gave very insightful comments, to which the authors have responded carrying out a great amount of additional work to address the questions raised. The addition of a full genome assembly of a male toad, and also the addition of an extensive transcriptomic analysis of tadpoles spanning most of the larval period of the species seems to have strengthened the support for the initial findings reported. At this stage, I think that the combined evidence derived from the chromosome-level genome assemblies provided, the transcriptomic analyses, the whole-genome pool-seq, ddRAD-seq and targeted sequencing are congruent enough to allow the authors to confidently describe the identified locus as a candidate sex determination one, only pending confirmation through downstream functional analyses. The authors seem to have also extensively revised the text from the previous version and seem to have addressed most of the previous comments. I therefore only have additional minor comments/suggestions to add at this point:

L. 61-62. Please use 'reptile' rather than 'reptilian vertebrate' and do not refer to poikilotherms as cold-blooded, as it is an old concept and rather inaccurate.

Response: We have replaced 'reptilian' by 'reptile' and deleted 'cold-blooded'.

L. 97-100. This sentence is a bit long and hard to follow. Would this other structure work better? 'Molecular evidence for male heterogamety resulted from sibship analyses using microsatellites and nuclear sequence markers (refs), identified the largest linkage group (LG1) as sex-linked in several diploid green toad species. This linkage group is homologous to the autosomal LG1 of *X. tropicalis* and harbours the gene *dmrt1*.'

Response: We have mostly adopted this proposal; it now reads: "Molecular evidence for male heterogamety from sibship analyses using microsatellites and nuclear sequence markers³³⁻³⁵, identified the largest linkage group (LG1) as sex-linked in several diploid green toad species. This linkage group is homologous to the autosomal LG1 of *X. tropicalis* and harbours the gene *dmrt1*."

L. 100-102. This sentence is not very clear either. Do you mean that all the green toad species show strongly reduced X/Y recombination but show some X/Y exchange in the pseudo-autosomal region over evolutionary timescales?

Response: Correct; we have adopted this proposal.

L. 115. The additional species included in the study are not revealed until page 24, at the end of the Methods section. I think it should be indicated in the introduction which taxa were studied, and most definitely at least when you report results on them (e.g. targeted enrichment analyses on L. 209).

Response: We followed this advice and list the other taxa now at the end of the Introduction.

Also, when referring to the Bufotes species you should use a recently updated species nomenclature, unless you have specific reasons not to do it. You indicate (L. 800) that you follow Ueda 1990 but perhaps it would be more appropriate to use the recent

Dufresnes, C., Mazepa, G., Jablonski, D., Oliveira, R. C., Wenseleers, T., Shabanov, D. A., ... & Litvinchuk, S. (2019). Fifteen shades of green: the evolution of Bufotes toads revisited. *Molecular Phylogenetics and Evolution*, 141, 106615.

Response: The ref. "32" in previous line 800 was a typo; it should be "38"; we have now corrected this. Indeed, follow the systematics of:

Betto-Colliard, C., Hofmann, S., Sermier, R., Perrin, N. & Stöck, M. Profound genetic divergence and asymmetric parental genome contributions as hallmarks of hybrid speciation in polyploid toads. *Proc Biol Sci* **285**, 20172667 (2018).

L. 117 X and Y chromosomes?

Response: Yes, plural corrected.

L. 139. The three distinct genetic families are not clearly described in the methods. You refer to one set of samples consisting of the two parent toads and 40 of their offspring, which I take as one genetic family, and then two groups of 25 or 15 siblings. Are these the two other genetic families? I take then that these two groups of siblings came from two additional pairs of adult toads? Please clarify.

Response: We know provided this information "(one comprising parents and 53 offspring, two other groups of 24 and 15 siblings from unknown parents)" in the text and and reworded this completely in the Methods.

L. 171. State the number and origin (e.g. tissue, larval or adult) of these transcriptomes and perhaps refer to the corresponding Methods section.

Response: We added to Transcriptome "of 54 larval, juvenile and adult toads (see below)".

L. 533. 'consisting of'. Also change to '40 offspring sexed mostly phenotypically'. How do you 'mostly' sex them phenotypically? Were some ambiguous? Did you use additional methods to sex them, other than phenotypically?

Response: We have updated this information for the individuals and reworded this completely.

L. 260. Delete extra parenthesis.

Response: Done.

L. 362. Our data on *B. viridis*...

Response: Corrected.

L. 497. Can you provide more detail as to the annotation procedure? Which RNA-Seq data were used? What annotation pipeline did you use?

Response: We added all the required information to Supplementary Text 1 and refer to this in the methods section.